# RISK-AWARE BAYESIAN REINFORCEMENT LEARNING FOR CAUTIOUS EXPLORATION

## ABSTRACT

This paper addresses the problem of maintaining safety during training in Reinforcement Learning (RL), such that the safety constraint violations are bounded at any point during learning. Whilst enforcing safety during training might limit the agent's exploration, we propose a new architecture that handles the trade-off between efficient progress in exploration and safety maintenance. As the agent's exploration progresses, we update Dirichlet-Categorical models of the transition probabilities of the Markov decision process that describes the agent's behaviour within the environment by means of Bayesian inference. We then propose a way to approximate moments of the agent's belief about the risk associated to the agent's behaviour originating from local action selection. We demonstrate that this approach can be easily coupled with RL, we provide rigorous theoretical guarantees, and we present experimental results to showcase the performance of the overall architecture.

## 1 INTRODUCTION

Traditionally, RL is principally concerned with the policy that the agent generates by the end of the learning process. In other words, the agent's policy *during* learning is overlooked to the benefit of learning how to behave optimally. Accordingly, many standard RL methods rely on the assumption that the agent selects each available action at every state infinitely often during exploration (Sutton et al., 2018; Puterman, 2014). A related technical assumption that is often made is that the MDP is *ergodic*, meaning that every state is reachable from every other state under proper action selection (Moldovan & Abbeel, 2012). These assumptions may sometimes be reasonable, e.g., in virtual environments where restarting is always an option. However, in safety-critical systems, these assumptions might be unreasonable, as we may explicitly require the agent to never visit certain unsafe states. Indeed, in a variety of RL applications the safety of the agent is particularly important, e.g. expensive autonomous platforms or robots that work in proximity of humans. Thus, researchers are recently paying increasing attention not only to maximising a long-term task-driven reward, but also to enforcing avoidance of unsafe training.

**Related Work** The general problem of *Safe* RL has been an active area of research in which numerous approaches and definitions of safety have been proposed (Brunke et al., 2021; Garcia & Fernandez, 2015; Pecka & Svoboda, 2014). In (Moldovan & Abbeel, 2012), safety is defined in terms of ergodicity, with the goal of safety being that an agent is always able to return to its current state after moving away from it. In (Chow et al., 2018a), safety is pursued by minimising a cost associated with worst-case scenarios, when cost is associated with a lack of safety. Similarly, (Miryoosefi et al., 2019) defines the safety constraint in terms of the expected sum of a vector of measurements to be in a target set. Other approaches (Li & Belta, 2019; Hasanbeig et al., 2019a;b; 2020; Cai et al., 2021; Hasanbeig et al., 2022) define safety by the satisfaction of temporal logical formulae of the learnt policy, but do not provide safety *while* training such policy. Many existing approaches have been concerned with providing guarantees on the safety of the learned policy sometimes under the assumption that a backup policy is available (Coraluppi & Marcus, 1999; Perkins & Barto, 2002; Geibel & Wysotzki, 2005; Mannucci et al., 2017; Chow et al., 2018b; Mao et al., 2019). These methods are applicable to systems if they can be trained on accurate simulations, but for many other real-world systems we instead require safety *during* training.

There has also been much research done into the development of approaches to maintaining safety during training. For instance, (Alshiekh et al., 2017; Jansen et al., 2019; Giacobbe et al., 2021) leverage the concept of a *shield* that stops the agent from choosing any unsafe actions. The shield

assumes the agent has to observe the entire MDP (and opponents) to construct a safety (game) model, which will be unavailable for many partially-known MDP tasks. The approach in (Garcia & Fernandez, 2012) assumes a predefined safe baseline policy that is most likely sub-optimal, and attempts to slowly improve it with a slightly noisy action-selection policy, while defaulting to the baseline policy whenever a measure of safety is exceeded. However, this measure of safety assumes that nearby states have similar safety levels, which may not always be the case. Another common approach is to use expert demonstrations to attempt to learn how to behave safely (Abbeel et al., 2010), or even to include an option to default to an expert when the risk is too high (Torrey & Taylor, 2012). Obviously, such approaches rely heavily on the presence and help of an expert, which cannot always be counted upon. Other approaches on this problem (Wen & Topcu, 2018; Cheng et al., 2019; Turchetta et al., 2016) are either computationally expensive or require explicit, strong assumptions about the model of agent-environment interactions. Crucially, maintaining safety in RL by efficiently leveraging available data is an open problem (Taylor et al., 2021).

**Contributions** We tackle the problem of synthesising a policy via RL that optimises a discounted reward, while not violating a safety requirement *during* learning. This paper puts forward a *cautious RL scheme* that assumes the agent maintains a Dirichlet-Categorical model of the MDP. We incorporate higher-order information from the Dirichlet distributions, in particular we compute approximations of the (co)variances of the risk terms. This allows the agent to reason about the contribution of epistemic uncertainty to the risk level, and therefore to make better informed decisions about how to stay safe during learning. We show convergence results for these approximations, and propose a novel method to derive an approximate bound on the confidence that the risk is below a certain level. The new method adds a functionality to the agent that prevents it from taking critically risky actions, and instead leads the agent to take safer actions whenever possible, but otherwise leaves the agent to explore as normal. The proposed method is versatile given that it can be added on to general RL training schemes, in order to maintain safety during learning.

## 2 BACKGROUND

### 2.1 PROBLEM SETUP

**Definition 2.1** *A finite MDP with rewards (Sutton et al., 2018) is a tuple $M = \langle Q, A, q_0, P, Re \rangle$ where $Q = \{q^1, q^2, q^3, ..., q^N\}$ is a finite set of states, $A$ is a finite set of actions, without loss of generality $q_0$ is the initial state, $P(q'|q, a)$ is the probability of transitioning from state $q$ to state $q'$ after taking action $a$, and $Re(q, a)$ is a real-valued random variable which represents the reward obtained after taking action $a$ in state $q$. A realisation of this random variable (namely a sample, obtained for instance during exploration) will be denoted by $re(q, a)$.*

An agent is placed at $q_0 \in Q$ at time step $t = 0$. At every time step $t \in \mathbb{N}_0$, the agent selects an action $a_t \in A$, and the environment responds by moving the agent to some new state $q_{t+1}$ according to the transition probability distribution, i.e., $q_{t+1} \sim P(\cdot|q_t, a_t)$. The environment also assigns the agent a reward $re(q_t, a_t)$. The objective of the agent is to learn how to maximise the long term reward. In the following we explain these notions more formally.

**Definition 2.2** *A policy $\pi$ assigns a distribution over $A$ at each state: $\pi(a|q)$ is the probability of selecting action $a$ in state $q$. Given a policy $\pi$, we can then define a state-value function*

$$v_\pi(q) = \mathbb{E}^\pi \left[ \sum_{t=0}^\infty \gamma^t re(q_t, a_t) \middle| q_0 = q \right],$$

*where $\mathbb{E}^\pi[\cdot]$ denotes the expected value given that actions are selected according to $\pi$, and $0 < \gamma \leq 1$ is a discount factor.*

Specifically, this means that the sequence $q_0, a_0, q_1, a_1, ...$ is such that $a_n \sim \pi(\cdot|q_n)$ and $q_{n+1} \sim P(\cdot|q_n, a_n)$. The discount factor $\gamma$ is a pre-determined hyper-parameter that causes immediate rewards to be worth more than rewards in the future, as well as ensuring that this sum is well-defined, provided the standard assumption of bounded rewards. The agent's goal is to learn an optimal policy, namely one that maximises the expected discounted return. This is actually equivalent to finding a policy that maximises the state-value function $v_\pi(q)$ at every state (Sutton et al., 2018).

**Definition 2.3** *A policy $\pi$ is optimal if, at every state $q$, $v_\pi(q) = v_*(q) = \max_{\pi'} v_{\pi'}(q)$.*

**Definition 2.4** *Given a policy $\pi$, we can define a state-action-value function $v_\pi(q, a) = \mathbb{E}^\pi \left[ \sum_{t=0}^\infty \gamma^t re(q_t, a_t) \mid q_0 = q, a_0 = a \right]$, similarly to the state-value function. This allows us to reinterpret the state-value function as $v_\pi(q) = \sum_a v_\pi(q, a)\pi(a|q)$, and thus we can see that an optimal deterministic policy $\pi$ must assign zero probability to any action $a$ that doesn't maximise the state-action value function.*

## 2.2 DIRICHLET-CATEGORICAL MODEL OF THE MDP

We consider a model for an MDP with unknown transition probabilities (Ghavamzadeh et al., 2015). The transition probabilities for a given state-action pair are assumed to be described by a categorical distribution over the next state. We maintain a Dirichlet distribution over the possible values of those transition probabilities: since the Dirichlet distribution is conjugate to the categorical distribution, we can employ Bayesian inference to update the Dirichlet distribution, as new observations are made while the agent explores the environment.

Formally, for each state-action pair $(q^i, a)$, we have a Dirichlet distribution $p_a^{i1}, p_a^{i2}, ..., p_a^{iN} \sim Dir(\alpha_a^{i1}, \alpha_a^{i2}, ...\alpha_a^{iN})$. The random variable $p_a^{ij}$ represents the agent's belief about the transition probability $P(q^j|q^i, a)$. At the start of learning, the agent will be assigned a prior Dirichlet distribution for each state-action pair, according to its initial belief about the transition probabilities. At every time step, as the agent moves from some state $q^i$ to some state $q^k$ by taking action $a$, it will generate an event $q^i \xrightarrow{a} q^k$, which constitutes new data for the Bayesian inference. From Bayes' rule:

$$Pr(\mathbf{p_a^i} = \mathbf{x_a^i}|q^i \xrightarrow{a} q^k) \propto Pr(q^i \xrightarrow{a} q^k|\mathbf{p_a^i} = \mathbf{x_a^i})Pr(\mathbf{p_a^i} = \mathbf{x_a^i})$$
$$= x_a^{ik}\prod_j (x_a^{ij})^{\alpha_a^{ij}-1} = [\prod_{j \neq k}(x_a^{ij})^{\alpha_a^{ij}-1}](x_a^{ik})^{(\alpha_a^{ik}+1)-1},$$

which immediately yields

$$Pr(\mathbf{p_a^i} = \mathbf{x_a^i}|q^i \xrightarrow{a} q^k) = Dir(\alpha_a^{i1}, \alpha_a^{i2}, ..., \alpha_a^{ik}+1, ..., \alpha_a^{iN}).$$

Thus, the posterior distribution is also a Dirichlet distribution. This update is repeated at each time step: the relevant information to the agent's posterior belief about the transition probabilities is the starting prior $Dir(\alpha_a^{i1}, \alpha_a^{i2}, ...\alpha_a^{iN})$ and the "transition counts" $c_a^{ij}$, keeping track of the number of times that $q^i \xrightarrow{a} q^j$ has occurred. The agent's posterior is then $(p_a^{i1}, p_a^{i2}, ..., p_a^{iN}) \sim Dir(\alpha_a^{i1}, \alpha_a^{i2}, ...\alpha_a^{iN})$: from this distribution, we can distill the expected value $\bar{p}_a^{ij}$ of each random variable $p_a^{ij}$, as well as the covariance of any two $p_a^{ij}$ and $p_a^{ik}$ (therefore also the variance of a single $p_a^{ij}$):

$$\bar{p}_a^{ij} = \mathbb{E}[p_a^{ij}] = \frac{\alpha_a^{ij}}{\alpha_a^{i0}}, \qquad Cov[p_a^{ij}, p_a^{ik}] = \frac{\alpha_a^{ij}(\delta^{jk}\alpha_a^{i0} - \alpha_a^{ik})}{(\alpha_a^{i0})^2(\alpha_a^{i0}+1)},$$

where $\alpha_a^{i0} = \sum_{k=1}^N \alpha_a^{ik}$, and $\delta^{jk}$ is the Kronecker delta.

## 3 RISK-AWARE BAYESIAN RL FOR CAUTIOUS EXPLORATION

In this section we propose a new approach to Safe RL, which will specifically address the problem of how to learn an optimal policy in an MDP with rewards, while avoiding certain states classified as unsafe during training. The agent is assumed to know which states of the MDP are safe and which are unsafe, but instead of assuming that the agent has this information globally, namely for all states of the MDP, we find it more reasonable that the agent observes states within an area around itself. This closely resembles real-world situations, where systems may have sensors that allow them to detect close-by danger areas, but not necessarily know about danger zones that are far away from them. In particular, we assume that there is an observation "boundary" $O$, such that the agent can observe all states that are reachable from the current state within $O$ steps and distinguish which of those states are safe or unsafe. The rest of this section is structured as follows:

1. In Section 3.1, we define the risk $r_c^m(a)$ over $m$ steps of taking an action $a$ at the current state, denoted as $q^c$. We then introduce a random variable $R_c^m(a)$ representing the agent's belief about the risk;

2. In Section 3.2, we leverage a method from (Casella & Berger, 2021) to approximate the expected value and variance of the random variable $R_c^m(a)$. We provide convergence results on the approximations of the expectation and variance of $R_c^m(a)$;

3. In Section 3.3, we show how the Cantelli Inequality (Cantelli, 1929) allows us to estimate a confidence bound on the risk $r_c^m(a)$;

4. In Section 3.4, we prescribe a methodology for incorporating the expectation and variance of risk into the action selection during the training of an RL agent.

## 3.1 DEFINITION AND CHARACTERISATION OF THE RISK

Given the observation boundary $O$, we reason about the risk incurred over the next $m$ steps after taking a particular action $a$ in the current state $q^c$, for any $m \leq O$. However, note that there is a dependence between the agent's estimate of such a risk and the use of that estimate to inform its action selection policy. In order to solve this dilemma we severe the dependency between the risk that we calculate and the actions selected generating that risk by fixing a policy over the $m$-step horizon, and calculating the risk given that policy. Similar to temporal-difference learning schemes, this is done by assuming best-case action selection, namely, the $m$-step risk $r_c^m(a)$ at state $q^c$ after taking action $a$ is defined assuming that after selecting action $a$, the agent will select subsequent actions to minimize the expected risk going forward. Assuming that the agent is at state $q^c$, we define the agent's approximation of the $m$-step risk $\bar{R}_c^m(a)$ by back-propagating the risk given the "expected safest policy" over $m$ steps, as follows:

$$\bar{R}_k^0 = \mathbb{1}(q^k \text{ is observed and unsafe}); \tag{1}$$

$$\bar{R}_k^{n+1}(a) = \begin{cases} 1 & \text{if } q^k \text{ is observed and unsafe} \\ \sum_{j=1}^N \bar{p}_a^{kj} \bar{R}_j^n & \text{otherwise}; \end{cases} \tag{2}$$

$$\bar{R}_k^{n+1} = \begin{cases} 1 & \text{if } q^k \text{ is observed and unsafe} \\ \min_{a \in A} \bar{R}_k^{n+1}(a) & \text{otherwise}. \end{cases} \tag{3}$$

We terminate this iterative process at $n+1 = m$ and once we have calculated $\bar{R}_c^m(a)$ ($c = k$) for all actions $a$. Note that, despite the use of progressing indices $n$, this is an iterative back-propagation that leverages the expected values of agent's belief about the transition probabilities, i.e., $\bar{p}_a^{kj}$. Thus, $\bar{R}_c^m(a)$ is the agent's approximation of the expectation of the probability of entering an unsafe state within $m$ steps by selecting action $a$ at state $q^c$, and thereafter by selecting actions that it currently believes will minimize the probability of entering unsafe states over the given time horizon.

The term $\bar{p}_a^{kj} = \mathbb{E}[p_a^{kj}]$ is used as a point estimate of the true transition probability $t_a^{kj} = P(q^j|q^k, a)$. The value of $\bar{R}_c^m$ only relies on states which the agent believes are reachable from $q^c$ within $m$ steps. In particular so long as the horizon $m$ is less than the observation boundary $O$, the agent is able to observe all states which are relevant to the calculation of $\bar{R}_c^m(a)$, so specifically, $\mathbb{1}(q^j \text{ is unsafe}) = \mathbb{1}(q^j \text{ is observed and unsafe})$ for all relevant states $q^j$ (see Appendix G for more details).

## 3.2 APPROXIMATION OF EXPECTED VALUE AND COVARIANCE OF THE RISK

Let $\mathbf{x}$ denote the vector of variables $x_a^{ij}$ where $i, j$ range from 1 to $N$ and $a$ ranges over $A$, i.e., $\mathbf{x} = \left((x_a^{ij})_{i,j=1,\dots,N \text{ and } a \in A}\right)$. We assume that these indices are ordered lexicographically by $(i, a, j)$. This is because $i$ and $a$ will be used to signify a state-action pair $(q^i, a)$, and $j$ will be used to signify a potential next state $q^j$. Introduce a set of functions (we shall see they take the shape of polynomials) $g_k^n[\mathbf{x}]$ defined, for each state $q^k$, as follows:

$$g_k^0[\mathbf{x}] := \mathbb{1}(q^k \text{ is observed and unsafe});$$

$$g_k^{n+1}(a)[\mathbf{x}] := \begin{cases} 1 & \text{if } q^k \text{ is observed and unsafe} \\ \sum_{j=1}^N x_a^{kj} g_j^n[\mathbf{x}] & \text{otherwise}; \end{cases}$$

$$g_k^{n+1}[\mathbf{x}] := \begin{cases} 1 & \text{if } q^k \text{ is observed and unsafe} \\ g_k^{n+1}\left(\arg\min_a \bar{R}_k^{n+1}(a)\right)[\mathbf{x}] & \text{otherwise}. \end{cases}$$

Then we can write the risk (of selecting action $a$ in state $q^c$, over $m$ steps) defined above as $r_c^m(a) = g_c^m(a)[\mathbf{t}]$, where $\mathbf{t} = \left((t_a^{ij})_{i,j=1,\dots,N \text{ and } \forall a \in A}\right)$ is a vector of all "true" transition probabilities $t_a^{ij} := P(q^j|q^i, a)$. We can similarly write the agent's approximation of the risk as $\bar{R}_c^m(a) = g_c^m(a)[\bar{\mathbf{p}}]$, where similarly $\bar{\mathbf{p}} = \left((\bar{p}_a^{ij})_{i,j=1,\dots,N \text{ and } a \in A}\right)$. We refer to the actions specified by these argmin operators as the *agent's expected safest action* in each state over the next

$m$ steps. Now, crucially, we can also define a new random variable $R_c^m(a) = g_c^m(a)[\mathbf{p}]$, where $\mathbf{p} = \big((p_a^{ij})_{i,j=1,\ldots,N \text{ and } \forall a \in A}\big)$. Since the $p_a^{ij}$s are random variables representing the agent's beliefs about the true transition probabilities $t_a^{ij}$, we in fact have that this random variable $R_c^m(a)$ represents the agent's beliefs about the true risk $r_c^m(a)$. In the following, we show that $\bar{R}_c^m(a)$ can be viewed as an approximation of $\mathbb{E}[R_c^m(a)]$, and we provide and justify an approximation of $Var[R_c^m(a)]$. These approximations can be used by the agent to reason about the true risk $r_c^m(a)$.

In order to construct approximations of the expectation and variance of $R_c^m(a)$, we make use of the first-order Taylor expansion of $g_c^m(a)[\mathbf{x}]$ around $\mathbf{x} = \bar{\mathbf{p}}$, following a method in (Casella & Berger, 2021). The Taylor expansion is

$$g_c^m(a)[\mathbf{x}] = g_c^m(a)[\bar{\mathbf{p}}] + \sum_{i,j=1}^{N} \sum_{b \in A} \frac{\partial g_c^m(a)}{\partial x_b^{ij}}(x_b^{ij} - \bar{p}_b^{ij}) + \text{ remainder}, \tag{4}$$

where the partial derivatives are also evaluated at $\bar{\mathbf{p}}$. Now we can turn equation 4 into a statistical approximation by dropping the remainder and reasoning over the random variables $\mathbf{p}$ for $\mathbf{x}$, namely:

$$g_c^m(a)[\mathbf{p}] \approx g_c^m(a)[\bar{\mathbf{p}}] + \sum_{i,j=1}^{N} \sum_{b \in A} \frac{\partial g_c^m(a)}{\partial x_b^{ij}}(p_b^{ij} - \bar{p}_b^{ij}). \tag{5}$$

We can then take the expectation of both sides, obtaining

$$\mathbb{E}[g_c^m(a)[\mathbf{p}]] \approx \mathbb{E}[g_c^m(a)[\bar{\mathbf{p}}]] + \mathbb{E}\big[\sum_{i,j=1}^{N} \sum_{b \in A} \frac{\partial g_c^m(a)}{\partial x_b^{ij}}(p_b^{ij} - \bar{p}_b^{ij})\big]$$

$$= g_c^m(a)[\bar{\mathbf{p}}] + \sum_{i,j=1}^{N} \sum_{b \in A} \frac{\partial g_c^m(a)}{\partial x_b^{ij}}\mathbb{E}[(p_b^{ij} - \bar{p}_b^{ij})] = g_c^m(a)[\bar{\mathbf{p}}],$$

where the above steps follow since the only random term in the right-hand side is $p_b^{ij}$, for which $\mathbb{E}(p_b^{ij}) = \bar{p}_b^{ij}$. Recall that $g_c^m(a)[\mathbf{p}] = R_c^m(a)$ and $g_c^m(a)[\bar{\mathbf{p}}] = \bar{R}_c^m(a)$. Thus, we now have $\bar{R}_c^m(a)$ as an approximation of the expectation of $R_c^m(a)$. For the approximation of the variance of the agent's believed risk, which is again a random variable, we can write:

$$Var(g_c^m(a)[\mathbf{p}]) \approx \mathbb{E}[(g_c^m(a)[\mathbf{p}] - g_c^m(a)[\bar{\mathbf{p}}])^2]$$

$$\approx \mathbb{E}\left[\left(\sum_{i,j=1}^{N} \sum_{b \in A} \frac{\partial g_c^m(a)}{\partial x_b^{ij}}(p_b^{ij} - \bar{p}_b^{ij})\right)^2\right] \quad \text{(from equation 5)}$$

$$= \sum_{i,j,s,t=1}^{N} \sum_{b_1,b_2 \in A} \frac{\partial g_c^m(a)}{\partial x_{b_1}^{ij}} \frac{\partial g_c^m(a)}{\partial x_{b_2}^{st}} Cov(p_{b_1}^{ij}, p_{b_2}^{st})$$

$$= \sum_{i=1}^{N} \sum_{b \in A} \sum_{j,t=1}^{N} \frac{\partial g_c^m(a)}{\partial x_b^{ij}} \frac{\partial g_c^m(a)}{\partial x_b^{it}} Cov(p_b^{ij}, p_b^{it}) = \bar{V}_c^m(a), \tag{6}$$

where $\bar{V}_c^m(a)$ is the approximation for the variance of $R_c^m(a)$, i.e., $\approx Var(R_c^m(a))$, and the last line follows from the fact that the covariance between two transition probability beliefs $p_{b_1}^{ij}$ and $p_{b_2}^{st}$ is always 0, unless they correspond to the same starting state-action pair $(q^i, b)$. In other words, $Cov(p_{b_1}^{ij}, p_{b_2}^{st}) = 0$ unless $i = j$ and $b_1 = b_2$. Next, we show consistency of the estimate in the limit (see Appendix E for the proof).

**Theorem 3.1** *Under Q-learning convergence assumptions (Watkins, 1989), namely that reachable state-action pairs are visited infinitely often, the estimate of the mean of the believed risk distribution $\bar{R}_c^m(a)$ converges to the true risk $r_c^m(a)$, and it does so with the variance of the believed risk distribution $Var(g_c^m(a)[\mathbf{p}])$ approaching the estimate of that variance $\bar{V}_c^m(a)$. Specifically,*

$$\frac{(\bar{R}_c^m(a) - r_c^m(a))}{\sqrt{\bar{V}_c^m(a)}} \to \mathcal{N}(0,1) \text{ in distribution.}$$

### 3.3 Estimating a Confidence on the Approximation of the Risk

So far we have shown that when the agent is in the current state $q^c$, for each possible action $a$, approximations of the expectation and variance of its belief $R_c^m(a)$ about the risk $r_c^m(a)$ can be formally obtained: we denote these two approximations by $\bar{R}_c^m(a)$ and $\bar{V}_c^m(a)$, respectively. We describe a method for combining these approximations to obtain a bound on the level of confidence that the risk $r_c^m(a)$ is below a certain threshold.

We appeal to the Cantelli Inequality, which is a one-sided Chebychev bound (Cantelli, 1929). Having computed $\bar{R}_c^m(a)$ and $\bar{V}_c^m(a)$, for a particular confidence value $0 < C < 1$ we can define $P_a := \bar{R}_c^m(a) + \sqrt{\frac{\bar{V}_c^m(a)C}{1-C}}$. From the Cantelli Inequality we then have

$$\Pr(R_c^m(a) \leq P_a) \geq C.$$

Specifically, $P_a$ is the lowest risk level such that, according to its approximations, the agent can be at least $100 \times C$ % confident that the true risk is below level $P_a$. The agent can therefore leverage $P_a$ when attempting to perform safe exploration (please refer to Appendix F for more details).

### 3.4 Risk-aware Bayesian RL for Cautious Exploration (RCRL)

We propose a setup for Safe RL that leverages the expectation and variance of the risk to allow an agent to explore the environment safely, while attempting to learn an optimal policy. In order to pick the most optimal yet safe action at each state, we propose a *double-learner* architecture, referred to as *Risk-aware Cautious RL (RCRL)* and explained next.

The first learner is an optimistic agent that employs Q-learning (QL) to maximize the expected cumulative return. The second learner is a pessimistic agent that maintains a Dirichlet-Categorical model of the transition probabilities of the MDP. In particular, this agent is initialized with a prior $Pri$ that encodes any information the agent might have about the transition probabilities. For each state-action pair $(q^i, a)$ we have a Dirichlet distribution $p_a^{i1}, p_a^{i2}, ..., p_a^{iN} \sim Dir(\alpha_a^{i1}, \alpha_a^{i2}, ...\alpha_a^{iN})$. As the agent explores the environment, the Dirichlet distributions are updated using Bayesian inference.

For each action $a$ available in the current state $q^c$, the pessimistic learner computes the approximations $\bar{R}_c^m(a)$ and $\bar{V}_c^m(a)$ of its belief $R_c^m(a)$ of the risk over the next $m$ steps of taking action $a$ in $q^c$. The "risk horizon" $m$ is a hyper-parameter that, as discussed, should be set at most as the observation boundary $O$. The pessimistic learner is also initialized with two hyper-parameters $P_{\max}$ and $C(n)$: $P_{\max}$ represents the maximum level of risk that the agent should be prepared to take, whereas $C(n)$ is a decreasing function of the number of times $n$ that the current state has been visited, which satisfies $C(0) < 1$ and $\lim_{n \to \infty} C(n) = 0$. From Section 3.3, the agent can then compute, for each action $a$, the value

$$P_a = \bar{R}_c^m(a) + \sqrt{\frac{\bar{V}_c^m(a)C(n)}{1 - C(n)}}, \tag{7}$$

which can thus define a set of safe actions: these are all the actions that the agent believes have risk less than $P_{max}$, with confidence at least $C(n)$, namely

$$A_{safe} = \{a \in A | P_a \leq P_{max}\}.$$

In case there are no actions $a$ such that $P_a \leq P_{max}$, the agent instead allows

$$A_{safe} = \{a \in A | \bar{R}_c^m(a) = \min_{a'} \bar{R}_c^m(a)\}. \tag{8}$$

Finally, the agent selects an action $a^*$ from the set of safe actions using softmax action selection (Sutton et al., 2018) according to the Q-values of those actions, with some *temperature* $t > 0$:

$$Pr(a^* = a) = \frac{e^{Q(q^c, a)/t}}{\sum_{a \in A_{\text{safe}}} e^{Q(q^c, a)/t}}. \tag{9}$$

The pseudo-code for the full algorithm is available in Appendix B.

In summary, we effectively have two agents learning to accomplish two tasks. The first agent performs Q-learning to learn an optimal policy for the reward. The second agent determines the best approximation of the expected value and variance of each action, enabling it to prevent the first agent from selecting actions that it cannot guarantee to be safe enough (with at least a given confidence). When instead the pessimistic agent cannot guarantee that any action is safe enough, it forces

Table 1: Total successes and failures. Gridworld: different priors and acceptable risks $P_{\max}$, averaged over 10 agents. PacMan: varying risk horizon $m$, single agent.

| Experiment | Setup | # Successes | # Failures | Total Episodes |
|---|---|---|---|---|
| Gridworld | Prior 1, $P_{\max} = 0.33$ | 404.3 | 54.2 | 500 |
| | Prior 1, $P_{\max} = 0.01$ | 506.0 | 417.9 | 1500 |
| | Prior 2, $P_{\max} = 0.01$ | 384.6 | 0.5 | 500 |
| | Prior 3, $P_{\max} = 0.01$ | 407.4 | 14.4 | 500 |
| | Prior 3, $P_{\max} = 0.0033$ | 421.3 | 1.1 | 500 |
| | Native Q-Learning | 414.6 | 990.5 | 1500 |
| PacMan | Risk Horizon $m = 2$ | 234 | 77 | 311 |
| | Risk Horizon $m = 3$ | 207 | 68 | 275 |
| | Native Q-Learning | 0 | 1500 | 1500 |

the optimistic learner to go into "safety mode", i.e., to forcibly select the actions that minimize the expected value of the risk, as per equation 8. From an empirical perspective, implementing this concept of a "safety mode" allows for continued progress, and pairs extremely well with the definition of the risk: namely, when the agent deems that a state is too risky, it will go into this "safety mode" until it is back in a state with sufficiently safe actions.

Finally, note that $C(n)$ represents the level of confidence that the agent requires in an action being safe enough for it to consider taking that action. When the agent starts exploring and $C(n)$ is at its highest, the agent only explores actions that it is very confident in. However, it may need to take actions that it is less confident in order to find an optimal policy. Thus, as it continues exploring, $C(n)$ is reduced, allowing the agent to select actions upon which it is not as confident. However, in the limit, when $C(n) \to 0$, we have that $P_a = \bar{R}_c^m(a)$, which means that the agent never takes an action if its approximation of the expected value of the risk $\bar{R}_c^m(a)$ is more than the maximum allowable risk $P_{max}$.

## 4 EXPERIMENTS

**Gridworld -** We first evaluated the performance of RCRL on a *Slippery Gridworld Bridge* example. The states of the MDP consist of a $20 \times 20$-grid, as depicted in Figure 2a (Appendix C). The agent is initialized at $q_0$ in the bottom-left corner (green). The agent's task is to get to the goal region without ever entering an unsafe state. In particular, upon reaching a goal state, the agent is given a reward of 1 and the learning episode is terminated; at every other state it receives a reward of 0, and upon reaching an unsafe state the learning episode terminates with reward 0. At each time step the agent might move into one of the 4 neighbouring states, or stay in its current position; thus, the agent has access to 5 actions at each state, $A = \{right, up, left, down, stay\}$. If the agent selects action $a \in A$, then it has a 96% chance of moving in direction $a$, and a 4% chance of "slipping", namely moving to another random direction. If any movement would ever take the agent outside of the grid, then the agent will just remain in place. The agent is assumed to have an observation boundary $O = 2$ steps. Note that due to the slipperiness of the grid and the narrow passage to reach the goal state, minimizing the risk is not aligned with maximizing the expected reward.

We tested RCRL with 5 different combinations of a prior $Pri$ and a maximum acceptable risk $P_{\max}$. The following additional hyper-parameters of the algorithm were kept constant: the maximum number of steps per episode $max\_steps = 400$, the maximum number of episodes $max\_episodes = 500$ (although this was increased to 1500 in two cases when the agent did not converge to near-optimal policy within the first 500, cf. Table 1); the learning rate $\mu = 0.85$; the discount factor $\gamma = 0.9$; and the risk horizon $m = 2$ (Appendix B). Recall that a prior consists of a Dirichlet distribution $p_a^{i1}, ..., p_a^{iN} \sim Dir(\alpha_a^{i1}, ..., \alpha_a^{iN})$ for every state-action pair $(q^i, a)$. We considered three priors:

- Prior 1 - completely uninformative: in this case we assigned a value of 1 to every $\alpha$. This yields a distribution that is uniform over its support.

- Prior 2 - weakly informative: we assigned a value of 12 to the $\alpha$ corresponding to moving in the correct direction, and a value of 1 to all other $\alpha$'s. This gives a distribution in between Prior 1 and Prior 3 in both degree of bias and concentration.

- Prior 3 - highly informative: we assigned a value of 96 to the $\alpha$ corresponding to moving in the correct direction, and a value of 1 to all other $\alpha$'s. This gives a distribution that is highly concentrated, and for which the mean values of the transition probability random variables are the true transition probabilities of the MDP, and hence unbiased.

We tested the algorithm with all three priors and a maximum acceptable risk of $P_{\max} = 0.01$ and repeating each experiment 10 times to take averages. On average, the agent with the highly informative prior (Prior 3) entered unsafe states 14.4 times, and always converged to near-optimality within about 200 steps, successfully crossing the bridge 407.4 times. For the other 78.2 episodes, the agent reached the episode limit within crossing the bridge or entering an unsafe state. The agent with Prior 2 interestingly only entered unsafe states an average of 0.5 times per experiment, and converged to a near-optimal policy within about 300 episodes, successfully crossing the bridge 384.6 times. On the other hand, the agent with Prior 1 only crossed the bridge less than 30 times. We therefore increased the total number of episodes to 1500 and tried again, yet still over half the time it did not converge to a near-optimal policy (Appendix A).

We then tested Prior 1 with a more lenient maximum acceptable risk of $P_{max} = 0.33$, and found that the agent this time managed to converge to near-optimality within around 200 episodes, entering unsafe states 54.2 times and successfully crossing the bridge 404.3 times. We also tested Prior 3 with a stricter $P_{max} = 0.0033$ and found out that it entered unsafe states only 1.1 times and succeeded 421.3 times, converging to near-optimality within 150 episodes (Appendix A).

Finally, we tested native Q-learning, without any safe learning scheme. This native scheme had almost no successful crossings of the bridge in the first 500 episodes, so we ran it for 1500 episodes and found that it only converged to a near-optimal policy about half the time, on average entering unsafe states 990.5 times and successfully crossing the bridge 414.6 times.

Table 1 summarizes the number of successes and failures for each agent. To understand better the rate of convergence to near-optimality, Figure 1 (Appendix A) displays the number of steps taken by the agent to cross the bridge at every successful episode (it displays 400 if the agent never managed to cross the bridge) averaged over the 10 experiments. On each graph we display for comparison the theoretical least number of steps it could cross the bridge in, which is 22. Note that because the grid-world is slippery, even an optimal policy would have fluctuations above the 22-steps line.

**Discussion** The first result of note is how poorly Prior 1 performs with $P_{max} = 0.01$. It mostly fails to converge to near-optimal behaviour even with 1500 steps as can be seen in Figure 1b (Appendix A), in fact seeming to converge slower than native Q-learning. This occurs because the maximum allowable risk is set too low for the given prior. In particular, there are two main issues with this. The first issue is a type of degenerate behaviour specific to our algorithm and to the completely uninformative prior with overly strict $P_{max}$: given that the agent starts with no information on the transition probabilities, it is unable to tell which actions are safe and which are unsafe. In particular, with $P_{max}$ at 1%, the first time the agent arrives at any state $q^c$ from which it can observe some unsafe state, it immediately goes into safety mode as it judges that the risk of every action is above 1%. Since it has no information on which action is safest, it randomly selects an action (assuming the Q-values were initialized to 0). If that randomly-selected action does not take the agent closer to a risky state, then after updating the agent's beliefs about the transition probabilities for that action, it will believe that action is the safest one from that state. Thus every time it encounters that state again, it will *always* select that action, never attempting any other actions. This behaviour can be seen in Figure 2b (Appendix C). The state $(13, 1)$ has been visited significantly more often than any other state. This has occurred because the first time the agent encountered that state, it chose action $stay$, and as above, from then on always chose $stay$ in state $(13, 1)$. This would cause the agent to remain in $(13, 1)$ until it slipped off of that state.

The second issue with having such a strict $P_{\max}$ could involve any prior. In this case $P_{\max}$ is set so low that actions that may be optimal are simply never tested, as the agent's initial belief about those actions causes the expected risk associated with them to always be greater than $P_{\max}$. This should not be viewed as an undesirable consequence of the algorithm, but rather as the algorithm working as intended. With the maximum allowable risk level $P_{\max}$ set so low, the agent judges that certain actions are riskier than acceptable and therefore does not take them. However, this does raise a more general question about the nature of safe learning in general: ensuring safety while learning necessarily means avoiding actions we believe are too dangerous, so if we want any guarantees on safety, then we must accept that the agent may be unable to explore the entire state space.

The second result of note is that Prior 3 performs much less safely than Prior 2 does at $P_{max} = 0.01$. This seems counter intuitive at first, given that Prior 3 is more accurate and more confident than Prior 2. However, the explanation is quite simple. Prior 3 (initially) causes the agent's expected belief to correctly predict that there is only a $1\%$ chance of moving to an unsafe state on a particular step if the agent selects the action to move away from it. On the other hand, Prior 2 causes the agent's expected belief to predict there is a $6.25\%$ chance of this happening. Thus, Prior 3 (correctly) evaluates the risk of moving within 1 step of a risky state as much lower than Prior 2 does. It is likely that at some points in the experiments, the agent with Prior 3 chose to move within 1 step of an unsafe state where an agent starting with Prior 2 (with the same experiences) would have rejected that action as too risky. The agent with Prior 3 would then be at risk of slipping into an unsafe state. In Figure 2c and 2d (Appendix C), we can see exactly this happening, where Prior 3 regularly visits state $(13, 8)$, which is adjacent to the unsafe state $(12, 8)$. Prior 2 instead regularly moves one more state to the right before moving up to row 13, since $(12, 9)$ is safe.

Prior 3 with $P_{max} = 0.0033$ shows how we can make use of a highly accurate prior to guarantee even less risk, and in this case the agent almost never enters unsafe states, while converging faster than any other setup to near-optimality.

The final result is that the rate of convergence of the native Q-learning agent is much slower on this MDP than the other agents (excluding Prior 1 with the inappropriate $P_{max} = 0.01$). As in Figure 1 (Appendix A), Q-learning took between 300 and 1500 episodes to converge when it did, and occasionally failed to converge, compared to 150-300 episodes for the four other agents to converge in all 10 experiments. This was even the case for the agent with the completely uninformative prior, with $P_{max} = 0.33$. This is a key result: it shows that not only can RCRL keep the agent safe during learning when possible, it may also direct the agent to explore more fruitful areas of the state-space. In this case study in particular, the native Q-learning agent entered unsafe states so often initially that it took many episodes before it was able to access the bridge and find the reward at the other side. Conversely, since the safe agents mostly avoided "sinking" situations, they were able to explore much more of the state space on each episode.

**PacMan -** We also evaluated the performance of RCRL on a *PacMan* example. Figure 3a (Appendix D) depicts the initial state of the environment, where the agent (PacMan) must get to both yellow dots (food) without getting caught by the ghost. Note that because both the agent and the ghost move through the maze, the PacMan MDP has about 10 times more states than the Gridworld, and up to 5 times more possible next states at any given state. Upon picking up the second piece of food, the agent is given a reward of 1 and the learning episode stops. Every other state incurs a reward of 0 and if the ghost catches PacMan, the learning episode stops with reward 0. The agent has access to four actions at each state, $A = \{right, up, left, down\}$ and will move in the direction selected, or if that direction moves into a wall, then it will stay still. The ghost will with $90\%$ probability move in the direction that takes it closest to the agent's next location, and with $10\%$ probability will move in a random direction. For this setup, we assumed an observation boundary $O = 3$ and compared two values of the risk horizon, $m = 2, 3$. We therefore kept constant the other parameters and hyper-parameters: the learning rate $\mu = 0.85$; the discount factor $\gamma = 0.9$; the maximum number of steps per episode $max\_steps = 400$; the maximum acceptable risk $P_{max} = 0.33$; the prior, which we set to be a completely uninformative prior as in the Gridworld example; the maximum number of episodes, which we set as 1500 or the number of episodes before the total rate of successful episodes exceeded $75\%$.

As in Table 1, the agent with a risk horizon of $m = 2$ steps exceeded a success rate of 75% after 311 episodes, having failed 77 times. The agent with the larger risk horizon of $m = 3$ only took 275 steps to exceed that success rate, and only failed 68 times. Figures 3b-3c (Appendix D) display the number of steps taken by the agent to win (or 400 if they lose) for each agent, as well as the running average number of steps over the previous 50 episodes.

**Discussion** The improvement in performance from $m = 2$ to 3 is likely due to the increased foresight of the agent leading it to move away from excessively risky scenarios further in advance, potentially avoiding entering a state from which entering a dangerous state is unavoidable. However, it may also be simply due to the fact that increasing the risk horizon leads to an overall increase in risk estimates, which will naturally cause more actions to be considered too risky and may reduce the number of failures. In other words, we may have been in a situation where decreasing the maximum acceptable risk $P_{max}$ would have led to similar improvements, and the increase in risk horizon was behaving functionally more like a decrease in $P_{max}$. Both risk-aware agents compare very favourably against the Native Q-Learning agent, which did not succeed once in 1500 episodes.

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

APPENDICES

# A   APPENDIX A. GRIDWORLD: AVERAGE NUMBER OF STEPS TO SUCCEED

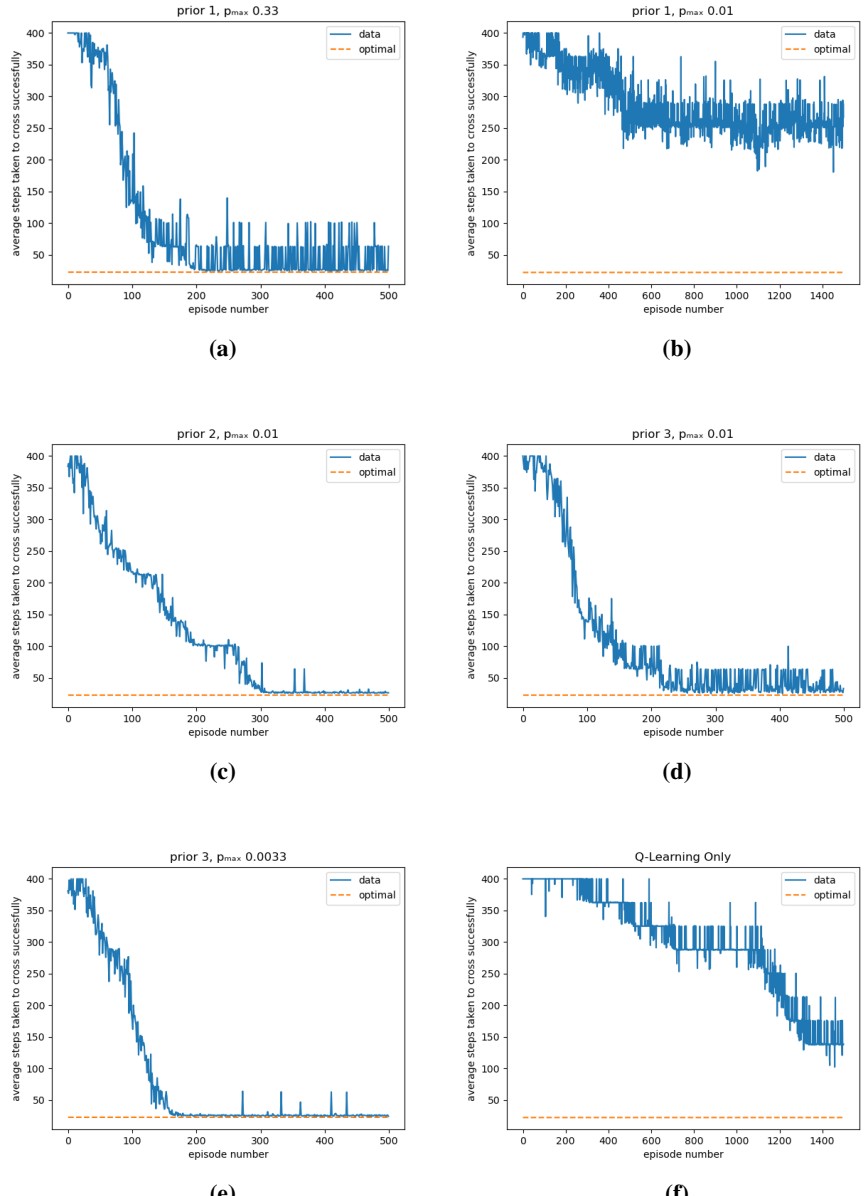

**Figure 1:** The number of steps it takes the agent to cross the bridge for every episode where it crosses. Averaged over 10 experiments. Results for Q-learning only and for RCRL across different priors and values of risk $P_{\max}$. As Q-learning converges, it approaches the lower bound on the optimal number of steps per episode.

## B   APPENDIX B. RISK-AWARE CAUTIOUS RL – PSEUDO CODE

---

**Algorithm 1:** Risk-aware Cautious RL (RCRL)

---

**input:** $Pri, C(n), P_{\max}, max\_steps, max\_episodes, \mu, \gamma, m$

(1) initialize $Q(q, a)$ for each state-action pair $(q, a)$;

(2) initialize $num\_steps = 0$ ;

(3) initialize $num\_episodes = 0$ ;

**while** $num\_episodes < max\_episodes$ **do**

(4)      $q^c \leftarrow q^0$;

(5)      $num\_episodes \leftarrow num\_episodes + 1$;

     **while** $num\_steps < max\_steps$ *and* $q^c$ *is not unsafe* **do**

(6)          calculate $\bar{R}_c^m(a)$ as in (2) ;

(7)          calculate $\bar{V}_c^m(a)$ as in (6) ;

(8)          calculate $P_a$ as in (7) ;

(9)          $A_{safe} := \{a \in A | P_a \leq P_{max}\}$ ;

         **if** $A_{safe} = \emptyset$ **then**

(10)             $A_{safe} \leftarrow \{a \in A | \bar{R}_c^m(a) = \min_{a'} \bar{R}_c^m(a)\}$ ;

         **end**

(11)          choose action $a^*$ according to (9) ;

(12)          pass action $a^*$ to environment and receive next state $q'$ and reward $re(q^c, a^*)$ ;

(13)          update belief $p$ as in section 2 ;

(14)          update $Q(q^c, a^*) \leftarrow (1 - \mu)Q(q^c, a^*) + \mu\left(re(q^c, a^*) + \gamma \max_{a'} Q(q', a')\right)$ ;

(15)          $q^c \leftarrow q'$ ;

(16)          $num\_steps \leftarrow num\_steps + 1$;

     **end**

**end**

---

# C Appendix C. Gridworld Experiment details

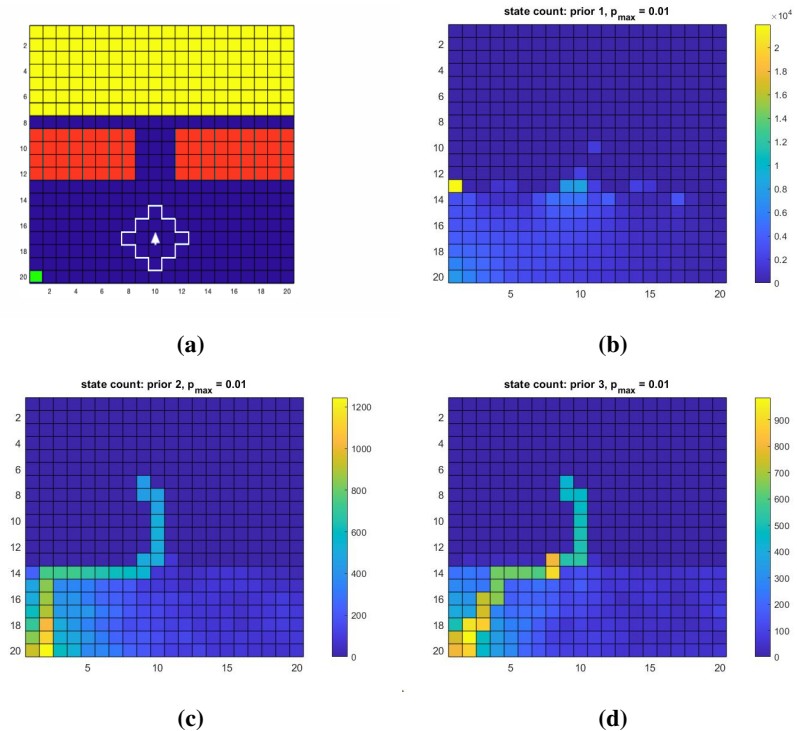

**Figure 2:** (a) Slippery Gridworld setup: agent is represented by an arrow surrounded by the observation area (white line). Labels denote target (yellow), unsafe (red) and safe states (blue), and initial state ($q_0$, green). (b) For a single experiment, number of state-visitations for Prior 1 at $P_{max} = 0.01$. (c-d) Number of state-visitations, for Priors 2 and 3 at $P_{max} = 0.01$.

# D    APPENDIX D. PACMAN EXPERIMENT DETAILS

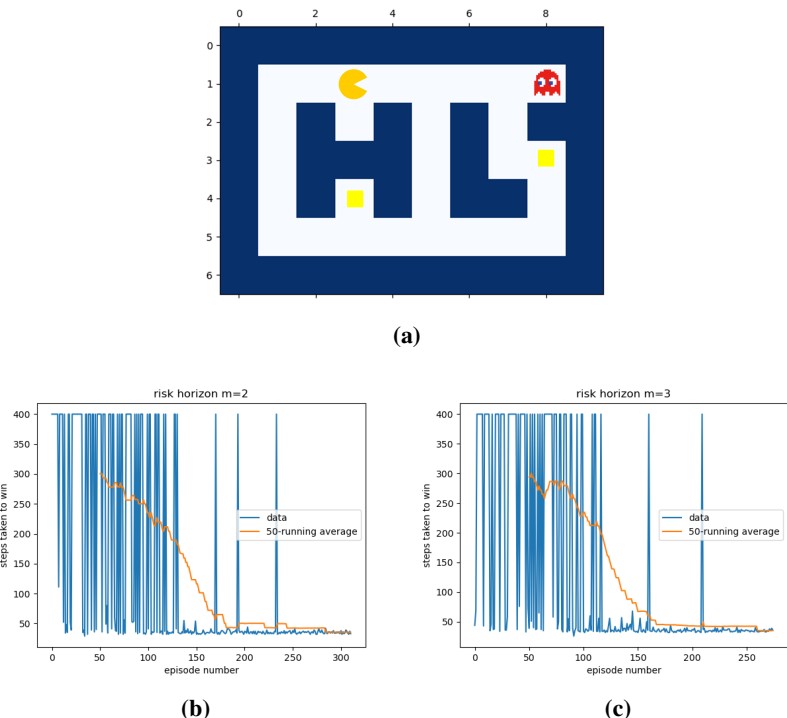

**(a)**

**(b)**                                                          **(c)**

**Figure 3:** (a) PacMan Setup: agent (PacMan) starts at position (1,3). Food is denoted by yellow dots, and the ghost starts in the top right corner. (b-c) Number of steps taken to win (i.e. eat both foods without being caught by the ghost) on episodes where the agent does win (or 400 if the agent is caught), for risk horizon 2 and 3. The orange line denotes the running average number of steps to win over the previous 50 episodes.

# E APPENDIX E. CONVERGENCE RESULTS FOR THE APPROXIMATIONS OF THE EXPECTED VALUE AND VARIANCE OF THE RISK

**Theorem E.1** *Under Q-learning convergence assumptions (Watkins, 1989), namely that reachable state-action pairs are visited infinitely often, the estimate of the mean of the believed risk distribution $\bar{R}_c^m(a)$ converges to the true risk $r_c^m(a)$, and it does so with the variance of the believed risk distribution $Var(g_c^m(a)[\mathbf{p}])$ approaching the estimate of that variance $\bar{V}_c^m(a)$. Specifically,*

$$\frac{\left(\bar{R}_c^m(a) - r_c^m(a)\right)}{\sqrt{\bar{V}_c^m(a)}} \to \mathcal{N}(0, 1) \text{ in distribution}$$

**Proof.**
Let us first rewrite the expressions in equation 6 in vector form, first introducing the following covariance matrix for $\mathbf{p}$:

$$\Sigma = \begin{pmatrix} Cov(p_{b_1}^{11}, p_{b_1}^{11}) & Cov(p_{b_1}^{11}, p_{b_1}^{12}) & \cdots & \\ Cov(p_{b_1}^{12}, p_{b_1}^{11}) & Cov(p_{b_1}^{12}, p_{b_1}^{12}) & & \\ \vdots & & \ddots & \\ & & & Cov(p_{b_M}^{NN}, p_{b_M}^{NN}) \end{pmatrix}.$$

Recall that the variables $p_a^{ij}$ are ordered lexicographically by $(i, a, j)$. Here we wrote $b_1$ for the first action in $A$ and $b_M$ for the last one, assuming $|A| = M$. Using matrix $\Sigma$, we can rewrite equation 6 for the approximate variance as

$$Var(R_c^m(a)) \approx (\nabla g_c^m(a)[\bar{p}])^T \Sigma (\nabla g_c^m(a)[\bar{p}]), \quad \nabla g_c^m(a)[\bar{p}] = \begin{bmatrix} \frac{\partial g_c^m(a)}{\partial x_{b_1}^{11}} \\ \frac{\partial g_c^m(a)}{\partial x_{b_1}^{12}} \\ \vdots \\ \frac{\partial g_c^m(a)}{\partial x_{b_M}^{NN}} \end{bmatrix}\Bigg|_{\mathbf{x}=\bar{\mathbf{p}}}, \quad (10)$$

where $\nabla g_c^m(a)[\bar{p}]$ is the gradient vector of $g_c^m(a)$ evaluated at $\bar{\mathbf{p}}$.

In the following, we employ the 'Delta Method' as described in (Casella & Berger, 2021) to allow us to derive a convergence result for the approximations for the mean and variance of $R_c^m(a)$ that we defined above. Let us introduce a semi-vectorised representation of equation 6 where we still leverage the fact that covariances across different state-action pairs are 0, i.e.,

$$\Sigma_b^i = \begin{pmatrix} Cov(p_b^{i1}, p_b^{i1}) & Cov(p_b^{i1}, p_b^{i2}) & \cdots & \\ Cov(p_b^{i2}, p_b^{i1}) & Cov(p_b^{i2}, p_b^{i2}) & & \\ \vdots & & \ddots & \\ & & & Cov(p_b^{iN}, p_b^{iN}) \end{pmatrix}$$

is the variance-covariance matrix for $\left((p_b^{ij})_{j=1,...,N}\right)$. Since $\Sigma$ is built by listing the $\Sigma_b^i$ along the diagonal for $i = 1, ..., N$ and $b \in A$, with zeros elsewhere, we have that equation 6 can be rewritten as

$$Var(R_c^m(a)) \approx \sum_{i=1}^{N} \sum_{b \in A} \left(\nabla_b^i g_c^m(a)[\bar{p}]\right)^T \Sigma \left(\nabla_b^i g_c^m(a)[\bar{p}]\right), \quad \nabla_b^i g_c^m(a)[\bar{p}] = \begin{bmatrix} \frac{\partial g_c^m(a)}{\partial x_b^{i1}} \\ \frac{\partial g_c^m(a)}{\partial x_b^{i2}} \\ \vdots \\ \frac{\partial g_c^m(a)}{\partial x_b^{iN}} \end{bmatrix}\Bigg|_{\mathbf{x}=\bar{\mathbf{p}}},$$

$$(11)$$

where $\nabla_b^i g_c^m(a)[\bar{p}]$ is the gradient vector $(\nabla g_c^m(a)[\bar{p}])$ restricted to entries $\frac{\partial g_c^m(a)}{\partial x_b^{ij}}$ for $j = 1, ..., N$.
We refer to this approximation for the variance of $R_c^m(a)$ as $\bar{V}_c^m(a)$ $(\approx Var(R_c^m(a)))$.
Consider the random vector $\mathbf{X} = (X_a^{ij})_{i,j=1,...,N \text{ and } a \in A}$ (with the previously discussed lexicographic order on the $X_a^{ij}$) where each $(X_a^{ij})_{j=1}^N$ follows a Categorical distribution with probabilities

$t_a^{ij}$ - i.e. a realisation of the vector $\mathbf{X}$ represents the result of taking one transition from every state-action pair. Wherever $X_a^{ij} = 1$ it represents a transition $q^i \xrightarrow{a} q^j$. $\mathbf{X}$ then has means $\mathbf{t}$ and covariances

$$Cov(X_a^{ij}, X_b^{st}) = \begin{cases} -t_a^{ij} t_b^{st} & \text{if } i = s \text{ and } a = b \\ 0 & \text{otherwise} \end{cases}$$

We can then write the variance-covariance matrix for $\mathbf{X}$ as

$$\Sigma_{\mathbf{XX}} = \begin{pmatrix} Cov(X_{b_1}^{11}, X_{b_1}^{11}) & Cov(X_{b_1}^{11}, X_{b_1}^{12}) & ... & \\ Cov(X_{b_1}^{12}, X_{b_1}^{11}) & Cov(X_{b_1}^{12}, X_{b_1}^{12}) & & \\ \vdots & & \ddots & \\ & & & Cov(X_{b_M}^{NN}, X_{b_M}^{NN}) \end{pmatrix},$$

If we observe independent random samples $\mathbf{X}^{(1)}, \mathbf{X}^{(2)}, ..., \mathbf{X}^{(n)}$ and denote the sample means as $\hat{X}_b^{ij} = \frac{1}{n} \sum_{k=1}^n (X_b^{ij})^{(k)}$, or $\hat{\mathbf{X}} = \frac{1}{n} \sum_{k=1}^n \mathbf{X}^{(k)}$ then for the function $g_c^n(a) [\mathbf{x}]$ we have,

$$g_c^m(a)[\hat{\mathbf{X}}] \approx g_c^m(a)[\mathbf{t}] + \sum_{i,j=1}^N \sum_{b \in A} \frac{\partial g_c^m(a)}{\partial x_b^{ij}} (\hat{X}_b^{ij} - t_b^{ij}),$$

This is a direct result from the first-order Taylor expansion around $\mathbf{t}$, and therefore the derivatives are evaluated at $\mathbf{t}$. In vector notation, we have

$$g_c^m(a)[\hat{\mathbf{X}}] \approx g_c^m(a)[\mathbf{t}] + (\nabla g_c^m(a)[\mathbf{t}])^T (\hat{\mathbf{X}} - t),$$

where

$$(\nabla g_c^m(a)[\mathbf{t}]) = \begin{bmatrix} \frac{\partial g_c^m(a)}{\partial x_b^{11}} \\ \frac{\partial g_c^m(a)}{\partial x_b^{12}} \\ \vdots \\ \frac{\partial g_c^m(a)}{\partial x_z^{NN}} \end{bmatrix} \Bigg|_{\mathbf{x}=\mathbf{t}}$$

From the 'Multivariate Delta Method' theorem (Casella & Berger, 2021), as long as

$$\tau^2 := (\nabla g_c^m(a)[\mathbf{t}])^T \Sigma_{\mathbf{XX}} (\nabla g_c^m(a)[\mathbf{t}]) > 0,$$

which we will prove later in Lemma 1 and Lemma 2, we have the following convergence:

$$\sqrt{n} \left( g_c^m(a)[\hat{\mathbf{X}}] - g_c^m(a)[\mathbf{t}] \right) \to \mathcal{N}(0, \tau^2) \text{ in distribution.} \tag{12}$$

Note that this is equivalent to

$$\frac{\sqrt{n} \left( g_c^m(a)[\hat{\mathbf{X}}] - g_c^m(a)[\mathbf{t}] \right)}{\tau} \to \mathcal{N}(0, 1) \text{ in distribution,} \tag{13}$$

where $\tau := \sqrt{\tau^2}$.

In the following we define $\bar{\mathbf{p}}^{(n)}$ and $\Sigma^{(n)}$ to be what $\bar{\mathbf{p}}$ and $\Sigma$ would have been had the agent started with it's prior about the transition probabilities $\mathbf{p}$ and then witnessed exactly the transitions represented by the random sample $\mathbf{X}^{(1)}, \mathbf{X}^{(2)}, ..., \mathbf{X}^{(n)}$. Formally, suppose that the agent's starting prior was, for each state-action pair $(q^i, b)$, that $p_b^{i1}, p_b^{i2}, ..., p_b^{iN} \sim Dir(\alpha_b^{i1}, \alpha_b^{i2}, ..., \alpha_b^{iN})$. Then we can consider the random variables $p_b^{i1(n)}, p_b^{i2(n)}, ..., p_b^{iN(n)} \sim Dir(\alpha_b^{i1} + n\hat{X}_b^{i1}, \alpha_b^{i2} + n\hat{X}_b^{i2}, ..., \alpha_b^{iN} + n\hat{X}_b^{iN})$. Since $n\hat{X}_b^{ij}$ is the count of the number of times $X_b^{ij}$ was 1 in the random sample, this new distribution is exactly the result of performing Bayesian inference on the prior given the random sample as our new data. We then let

$$\bar{p}_b^{ij(n)} := \mathbb{E}\left[ p_b^{ij(n)} \right] = \frac{\alpha_b^{ij} + n\hat{X}_b^{ij}}{\sum_{k=1}^N \left( \alpha_b^{ik} + n\hat{X}_b^{ik} \right)},$$

and we also define $\Sigma^{(n)}$ as the covariance matrix of the $p_b^{ij\,(n)}$ over all $i, j, b$, namely

$$\Sigma^{(n)} = \begin{pmatrix} Cov(p_b^{11(n)}, p_b^{11(n)}) & Cov(p_b^{11(n)}, p_b^{12(n)}) & \cdots & \\ Cov(p_b^{12(n)}, p_b^{11(n)}) & Cov(p_b^{12(n)}, p_b^{12(n)}) & & \\ \vdots & & \ddots & \\ & & & Cov(p_z^{NN(n)}, p_z^{NN(n)}) \end{pmatrix},$$

From Lemma 1, we have

$$\frac{\sqrt{n}\left(g_c^m(a)[\bar{\mathbf{p}}^{(n)}] - g_c^m(a)[\hat{\mathbf{X}}]\right)}{\tau} \to 0 \text{ in probability,} \tag{14}$$

and this allows us to use the well-known Slutsky's Theorem (Slutsky, 1925) on equation 14 and equation 13 to show that

$$\frac{\sqrt{n}\left(g_c^m(a)[\bar{\mathbf{p}}^{(n)}] - g_c^m(a)[\mathbf{t}]\right)}{\tau} \to \mathcal{N}(0, 1) \text{ in distribution.} \tag{15}$$

We must make one more modification to this result. Let

$$(\tau^{(n)})^2 := \left(\nabla g_c^m(a)[\bar{\mathbf{p}}^{(n)}]\right)^T \Sigma^{(n)} \left(\nabla g_c^m(a)[\bar{\mathbf{p}}^{(n)}]\right).$$

We would like to show that $n(\tau^{(n)})^2 \to \tau^2$ in probability. To do this, first note that $\bar{\mathbf{p}}^{(n)} \to \mathbf{t}$ in probability, so since $g_c^m(a)$ has continuous derivatives we have that $(\nabla g_c^m(a)[\bar{\mathbf{p}}^{(n)}]) \to (\nabla g_c^m(a)[\mathbf{t}])$ in probability. Next we note that $n\Sigma^{(n)} \to \Sigma_{\mathbf{XX}}$ in probability. This is because for the $(i, b_1, j), (s, b_2, t)$-entry we have $0 \to 0$ if $i \neq s$ or $b_1 \neq b_2$, and otherwise we have

$$\begin{aligned} nCov(p_b^{ij(n)}, p_b^{it(n)}) &= \frac{-n(\alpha_b^{ij} + n\hat{X}_b^{ij})(\alpha_b^{it} + n\hat{X}_b^{it})}{(\sum_{k=1}^N (\alpha_b^{ik} + n\hat{X}_b^{ik}))^2 (1 + \sum_{k=1}^N (\alpha_b^{ik} + n\hat{X}_b^{ik}))} \\ &= \frac{-n(\alpha_b^{ij} + n\hat{X}_b^{ij})(\alpha_b^{it} + n\hat{X}_b^{it})}{(n + \sum_{k=1}^N \alpha_b^{ik})^2 (n + 1 + \sum_{k=1}^N \alpha_b^{ik})} \\ &\to -t_b^{ij} t_b^{it} = Cov(X_b^{ij}, X_b^{it}). \end{aligned}$$

Therefore we have that the products converge in probability:

$$\begin{aligned} n(\tau^{(n)})^2 &= \left(\nabla g_c^m(a)[\bar{\mathbf{p}}^{(n)}]\right)^T n\Sigma^{(n)} \left(\nabla g_c^m(a)[\bar{\mathbf{p}}^{(n)}]\right) \\ &\to (\nabla g_c^m(a)[\mathbf{t}])^T \Sigma_{XX} (\nabla g_c^m(a)[\mathbf{t}]) = \tau^2. \end{aligned}$$

Since $\tau^2$ is always positive, and the square root function is therefore continuous at $\tau^2$, we have that $\sqrt{n}\tau^{(n)} \to \tau$, and so $\frac{\tau}{\sqrt{n}\tau^{(n)}} \to 1$ in probability. Now we can finally apply Slutsky's Theorem to obtain our final result, which is

$$\frac{\left(g_c^m(a)[\bar{\mathbf{p}}^{(n)}] - g_c^m(a)[\mathbf{t}]\right)}{\tau^{(n)}} \to \mathcal{N}(0, 1) \text{ in distribution.} \tag{16}$$

Recall that $g_c^m(a)[\mathbf{t}]$ is the actual risk in the current state $q^c$, $g_c^m(a)[\bar{\mathbf{p}}^{(n)}]$ is the agent's approximation of the expectation of the risk given it's beliefs, and $(\tau^{(n)})^2$ is the agent's approximation of the variance of the risk given it's beliefs (both, in this case, assuming it has seen exactly $n$ transitions from each state). So indeed our estimate of the mean of the believed risk distribution converges to the true risk with enough data, and it does so with the variance of the believed risk distribution approaching our estimate of that variance.

**Lemma 1** *Given the definition of the polynomial $g_c^m(a)[\mathbf{x}]$, we have the following:*

$$\frac{\sqrt{n}\left(g_c^m(a)[\bar{\mathbf{p}}^{(n)}] - g_c^m(a)[\hat{\mathbf{X}}]\right)}{\tau} \to 0 \text{ in probability}$$

**Proof.**
As required for the convergence results in Theorem 3.1, one can see that all of the coefficients in $g_c^m(a)[\mathbf{x}]$ are either 0 or 1. This means that we can rewrite it as a sum of terms of the form

$$\prod_{i,j,b}\left(x_b^{ij}\right)^{n_b^{ij}}$$

for exponents $n_b^{ij}$. This means that we can write

$$\frac{\sqrt{n}\left(g_c^m(a)[\bar{\mathbf{p}}^{(n)}] - g_c^m(a)[\hat{\mathbf{X}}]\right)}{\tau}$$

as a sum of terms of the form

$$\frac{\sqrt{n}}{\tau}\left(\prod_{i,j,b}\left(\bar{p}_b^{ij(n)}\right)^{n_b^{ij}} - \prod_{i,j,b}\left(\hat{X}_b^{ij}\right)^{n_b^{ij}}\right).$$

Substituting in the definition of $\bar{p}_b^{ij(n)}$ to this expression yields

$$\frac{\sqrt{n}}{\tau}\left(\prod_{i,j,b}\left(\frac{\alpha_b^{ij} + n\hat{X}_b^{ij}}{\sum_{k=1}^{N}\left(\alpha_b^{ik} + n\hat{X}_b^{ik}\right)}\right)^{n_b^{ij}} - \prod_{i,j,b}\left(\hat{X}_b^{ij}\right)^{n_b^{ij}}\right)$$

And we can simplify this by leveraging that $\sum_{k=1}^{N}\left(n\hat{X}_b^{ik}\right) = n$, to get

$$\frac{\sqrt{n}}{\tau}\left(\prod_{i,j,b}\left(\frac{\alpha_b^{ij} + n\hat{X}_b^{ij}}{n + \sum_{k=1}^{N}\alpha_b^{ik}}\right)^{n_b^{ij}} - \prod_{i,j,b}\left(\hat{X}_b^{ij}\right)^{n_b^{ij}}\right)$$

Now, the $\alpha_b^{ij}$ are constants, as is $\tau$, and the values of $\hat{X}_b^{ij}$ are all bounded between 0 and 1. Thus to show that this expression converges to 0 in probability, we will write it as one quotient, and show that some term in the denominator dominates all terms in the numerator. Let $M := \sum_{i,j,b} n_b^{ij}$. The expression above is equal to

$$\frac{\sqrt{n}}{\tau}\left(\frac{\prod_{i,j,b}\left(\alpha_b^{ij} + n\hat{X}_b^{ij}\right)^{n_b^{ij}} - \prod_{i,j,b}\left(\hat{X}_b^{ij}\left(n + \sum_{k=1}^{N}\alpha_b^{ik}\right)\right)^{n_b^{ij}}}{\prod_{i,j,b}\left(n + \sum_{k=1}^{N}\alpha_b^{ik}\right)^{n_b^{ij}}}\right)$$

Now on the numerator of the inner quotient, there are only two terms of order $n^M$. One is an

$$n^M\prod_{i,j,b}\left(\hat{X}_b^{ij}\right)^{n_b^{ij}}$$

that comes from the product on the left, and one is a

$$-n^M\prod_{i,j,b}\left(\hat{X}_b^{ij}\right)^{n_b^{ij}}$$

from the product on the right, and these cancel each other out. This means the numerator is entirely of order $n^{M-1}$ or less. On the other hand, the denominator of the inner quotient contains the term

$n^M$. Therefore, even after multiplying by the $\frac{\sqrt{n}}{\tau}$ on the outside, which would mean the highest order term on in the numerator could be as high as $n^{M-\frac{1}{2}}$, the $n^M$ in the denominator still dominates and the expression as a whole will converge to 0 in probability. Since

$$\frac{\sqrt{n}\left(g_c^m(a)[\bar{\mathbf{p}}^{(n)}] - g_c^m(a)[\hat{\mathbf{X}}]\right)}{\tau}$$

was a sum of expressions of that form, and they all converge to 0 in probability, we get the result we desired, which is that

$$\frac{\sqrt{n}\left(g_c^m(a)[\bar{\mathbf{p}}^{(n)}] - g_c^m(a)[\hat{\mathbf{X}}]\right)}{\tau} \to 0 \text{ in probability}$$

**Lemma 2** *The defined variable* $\tau^2 := (\nabla g_c^m(a)[\mathbf{t}])^T \Sigma_{\mathbf{XX}} (\nabla g_c^m(a)[\mathbf{t}])$ *is strictly greater than zero, namely* $\tau^2 > 0$.

**Proof.**
Note that the covariance matrix can be written as $\Sigma_{\mathbf{XX}} = \mathbb{E}[(\mathbf{X} - \mathbf{t})(\mathbf{X} - \mathbf{t})^T]$ (recall $\mathbf{t}$ is the mean vector for $\mathbf{X}$). So we have

$$\tau^2 = \mathbb{E}[(\nabla g_c^m(a)[\mathbf{t}])^T (\mathbf{X} - \mathbf{t})(\mathbf{X} - \mathbf{t})^T (\nabla g_c^m(a)[\mathbf{t}])$$
$$= \mathbb{E}[((\nabla g_c^m(a)[\mathbf{t}])^T (\mathbf{X} - \mathbf{t}))^2]$$

where we note that $s := (\nabla g_c^m(a)[\mathbf{t}])^T (\mathbf{X} - \mathbf{t})$ is a real-valued random variable, so $s^T = s$. Thus to prove $\tau^2 > 0$ we simply have to show that $s \neq 0$ for some value of $\mathbf{X}$ that occurs with non-zero probability.
Now,

$$s = \sum_{i,j,b} \left.\frac{\partial g_c^m(a)}{\partial x_b^{ij}}\right|_{\mathbf{x}=\mathbf{t}} (X_b^{ij} - t_b^{ij})$$

$$= \sum_{\text{state-action pairs } (q^i, b)} \left( \sum_{\text{possible next states } q^j} \left.\frac{\partial g_c^m(a)}{\partial x_b^{ij}}\right|_{\mathbf{x}=\mathbf{t}} (X_b^{ij} - t_b^{ij}) \right)$$

So let $s_b^i := \sum_{\text{states } q^j} \left.\frac{\partial g_c^m(a)}{\partial x_b^{ij}}\right|_{\mathbf{x}=\mathbf{t}} (X_b^{ij} - t_b^{ij})$, then $s = \sum_{\text{state-action pairs } (q^i, b)} s_b^i$.

We need to show that there is some possible value of $\mathbf{X}$ such that $s \neq 0$. Now the value of $\mathbf{X}$ is determined by the values of $\mathbf{X}_b^i := (X_b^{ij})_{j=1}^N$ for each state-action pair $(q^i, b)$. Furthermore, these $\mathbf{X}_b^i$ are independent, and the value of $s_b^i$ depends only on the value of $\mathbf{X}_b^i$. So if there is some state-action pair $(q^i, b)$ such that two possible values of $\mathbf{X}_b^i$ yield two distinct values of $s_b^i$ both with nonzero probability, then we can fix the values of the $X_{b'}^{hj}$ for all $j$ and all $(h, b') \neq (i, b)$ to be some values that occur with non-zero probability, which would fix the value of $s - s_b^i$, and so we could use our two distinct values of $s_b^i$ to find two distinct values of $s$. Both cannot be 0, so we would be done.

Now, the value of $\mathbf{X}_b^i$ is characterized by picking one $j$ s.t. $X_b^{ij} = 1$, and setting all other $X_b^{il} = 0$ for $l \neq j$. This means that to find two different values of some $s_b^i$, we just need to find states $q^i, q^j, q^l$ and an action $b$ such that the derivatives $\left.\frac{\partial g_c^m(a)}{\partial x_b^{ij}}\right|_{\mathbf{x}=\mathbf{t}}$ and $\left.\frac{\partial g_c^m(a)}{\partial x_b^{il}}\right|_{\mathbf{x}=\mathbf{t}}$ are distinct. Then setting $X_b^{ij} = 1$ would yield a different value of $s_b^i$ from setting $X_b^{il} = 1$. So long as the events $X_b^{ij} = 1$ and $X_b^{il} = 1$ both have nonzero probability, we would be done.

In order to show that such states $q^i, q^j, q^l$ and such an action $b$ exist, we must introduce vectors $A^n$ that will effectively keep track of each state's contribution towards $g_c^m(a)[\mathbf{t}]$ at the $n$th step of the risk backpropagation. First, define the $N$-by-$N$ matrix $P_n'[\mathbf{x}]$ for $n = 0, 1, ..., m - 2$ such that

$$(P_n'[\mathbf{x}])_{ij} = \begin{cases} 1 & \text{if } i = j \text{ and } q^i \text{ is unsafe and observed} \\ 0 & \text{if } i \neq j \text{ and } q^i \text{ is unsafe and observed} \\ x_{b_{in}}^{ij} & \text{otherwise} \end{cases}$$

where where $b_{in} := \arg\min_b \bar{R}_i^n(b)$. Define $P'_{m-1}[\mathbf{x}]$ as

$$(P'_{m-1}[\mathbf{x}])_{ij} = \begin{cases} 1 & \text{if } i = j \text{ and } q^i \text{ is unsafe and observed} \\ 0 & \text{if } i \neq j \text{ and } q^i \text{ is unsafe and observed} \\ x_a^{ij} & \text{otherwise} \end{cases}$$

Then the $P'_n[\mathbf{x}]$ represent the transition probabilities used in the calculation of $g_c^m(a)[\mathbf{x}]$. Specifically, we have that

- $g_k^n[\mathbf{x}]$ is the $k$th entry of the vector $(P'_{n-1}[\mathbf{x}])...(P'_0[\mathbf{x}])g^0$ for $n < m$

- $g_k^m(a)[\mathbf{x}]$ is the $k$th entry of the vector $(P'_{m-1}[\mathbf{x}])(P'_{m-2}[\mathbf{x}])...(P'_0[\mathbf{x}])g^0$

- So the risk at current state $q^c$, $g_c^m(a)[\mathbf{t}]$, is the $c$th entry of the vector $(P'_{m-1}[\mathbf{t}])(P'_{m-2}[\mathbf{t}])...(P'_0[\mathbf{t}])g^0$

where $g^0$ is the vector with entries $(g^0)_k := \mathbb{1}(q^k \text{ is observed and unsafe})$. We can now define the vectors $A^n$ for $n \leq m$ by

$$A_i^n := \begin{cases} \left((P'_{n-1}[\mathbf{t}])(P'_{n-2}[\mathbf{t}])...(P'_0[\mathbf{t}])g^0\right)_i & \text{if } q^i \text{ is safely reachable from } q^c \text{ in exactly } m - n \text{ steps} \\ 0 & \text{otherwise} \end{cases}$$

Where in this case a state $q^{s_n}$ is defined to be *safely reachable* from the current state $q^{s_0} = q^c$ in exactly $n$ steps if

- there are states $q^{s_1}, q^{s_2}, ..., q^{s_{n-1}}$ such that each $t_{b_{s_1}}^{s_p s_{p+1}} > 0$ for actions $b_{s_0} = a$ and $b_{s_k} := \arg\min_b \bar{R}_{s_p}^{m-k-1}(b)$ determined by the agent's expected safest policy, and

- the states $q^{s_1}, q^{s_2}, ..., q^{s_{n-1}}$ are all safe (note that $q^{s_n}$ can still be unsafe)

The purpose of these $A^n$ is just to restrict our attention to the states at step $n$ of the backpropagation that actually influence $g_c^m(a)[\mathbf{t}]$. It is easy to see that

$$\left((P'_{m-1}[\mathbf{t}])...(P'_n[\mathbf{t}])A^n\right)_c = g_c^m(a)[\mathbf{t}] \text{ for every } n = 0, 1, ..., m \tag{17}$$

Now we will be able to argue that if $g_c^m(a)[\mathbf{t}]$ is not equal to 0 or 1, there are states $q^i, q^j, q^l$ and an action $b$ such that $t_b^{ij}$ and $t_b^{il}$ are both non-zero (so there is a positive probability of the events $X_b^{ij} = 1$ and $X_b^{il} = 1$) and such that $\left.\frac{\partial g_c^m(a)}{\partial x_b^{ij}}\right|_{\mathbf{x}=\mathbf{t}} > \left.\frac{\partial g_c^m(a)}{\partial x_b^{il}}\right|_{\mathbf{x}=\mathbf{t}}$.

So assume that $g_c^m(a)[\mathbf{t}]$ is not equal to 0 or 1. Let $n_0$ be the largest index such that $A^{n_0}$ contains an entry $(A^{n_0})_l$ that is equal to 0 and such that $q^l$ is safely reachable from $q^c$ in exactly $m - n_0$ steps - so $(A^{n_0})_l$ is a 0 that came from $(P'_{m-1}[\mathbf{t}])((P'_{m-2}[\mathbf{t}])...(P'_0[\mathbf{t}])g^0)_l$.

Since $g_c^m(a)[\mathbf{t}]$ is not 0, $n_0 < m$, and since $q^l$ is safely reachable in $m - n_0$ steps, let $q^c = q^{s_0}, q^{s_1}, ..., q^{s_{m-n_0}} = q^l$ be a path along which $q^l$ is safely reachable. Then let $q^i = q^{s_{m-n_0-1}}$, and we have that $q^i$ is safe, and $t_{b_{s_{m-n_0-1}}}^{il} > 0$. For brevity, write $b' := b_{s_{m-n_0-1}}$

Now since $q^i$ is safely reachable in $m - (n_0 + 1)$ steps, $(A^{n_0+1})_i$ cannot be equal to 0 (since $n_0$ was maximal), so there must be some state $q^j$ such that $t_{b'}^{ij} > 0$ and $A_j^{n_0} > 0$, (in order for the term $t_{b'}^{ij} A_j^{n_0}$ to contribute some positive value to $A_i^{n_0+1}$). Finally, let $p$ be the probability of safely entering $q^i$ in $m - (n_0 + 1)$ steps (i.e., the sum over all paths that safely reach $q^i$ of the probability of taking that path by choosing the actions specified by the agent's expected safest policy). Then by the chain rule,

$$\left.\frac{\partial g_c^m(a)}{\partial x_{b'}^{ij}}\right|_{\mathbf{x}=\mathbf{t}} = p\left(1 \times A_j^{n_0} + t_{b'}^{ij} \times \left.\frac{\left((P'_{n_0-1}[\mathbf{x}])...(P'_0[\mathbf{x}])g^0\right)_j}{\partial x_{ij}}\right|_{\mathbf{x}=\mathbf{t}}\right) > 0$$

since clearly $\left.\frac{\left((P'_{n_0-1}[\mathbf{x}])...(P'_0[\mathbf{x}])g^0\right)_j}{\partial x_{ij}}\right|_{\mathbf{x}=\mathbf{t}}$ cannot be negative. On the other hand,

$$\frac{\partial g_c^m(a)}{\partial x_{b'}^{il}}\bigg|_{\mathbf{x}=\mathbf{t}} = p\left(1 \times (A^{n_0})_l + t_{b'}^{il} \times \frac{\left((P_{n_0-1}'[\mathbf{x}])...(P_0'[\mathbf{x}])g^0\right)_l}{\partial x_{b'}^{il}}\bigg|_{\mathbf{x}=\mathbf{t}}\right)$$

$$= p\left(1 \times 0 + t_{b'}^{il} \times \frac{\left((P_{n_0-1}'[\mathbf{x}])...(P_0'[\mathbf{x}])g^0\right)_l}{\partial x_{b'}^{il}}\bigg|_{\mathbf{x}=\mathbf{t}}\right) = 0$$

since only one of $t_{b'}^{il}$ and $\frac{\left((P_{n_0-1}'[\mathbf{x}])...(P_0'[\mathbf{x}])g^0\right)_l}{\partial x_{b'}^{il}}\bigg|_{\mathbf{x}=\mathbf{t}}$ can be nonzero - if increasing the value of $t_{b'}^{il}$ could increase the value of $(A^{n_0})_l = \left((P_{n_0-1}'[\mathbf{t}])...(P_0'[\mathbf{t}])g^0\right)_l$ from 0 to greater than 0, then $t_{b'}^{il}$ must have been 0 since $\left((P_{n_0-1}'[\mathbf{t}])...(P_0'[\mathbf{t}])g^0\right)_l$ is a sum of products of values from $\mathbf{t}$, all of which are non-negative.

Hence we have found states $q^i, q^j, q^l$ and an action $b'$ such that the derivatives $\frac{\partial g_c^m(a)}{\partial x_{b'}^{ij}}\bigg|_{\mathbf{x}=\mathbf{t}}$ and $\frac{\partial g_c^m(a)}{\partial x_{b'}^{il}}\bigg|_{\mathbf{x}=\mathbf{t}}$ are distinct. Hence the claim.

The only detail left to note is that we assumed that $g_c^m(a)[\mathbf{t}]$ is not either equal to 0 or 1. This assumption is reasonable to make, because if it did not hold, then either our agent would be doomed to enter an unsafe state within $m$ steps, or there is no chance of entering an unsafe state within $m$ steps, according to the agent's expected safest actions. Since what matters to us is how the agent manages risk, situations involving risk 1 or risk 0 are irrelevant.

## F  APPENDIX F. CONFIDENCE BOUND ON THE RISK

To estimate a confidence bound on the risk, we appeal to the Cantelli Inequality, which is a one-sided Chebychev bound (Cantelli, 1929), and states that for a real-valued random variable $R$ with expectation $\mathbb{E}[R]$ and variance $Var[R]$, for $\lambda > 0$ we have

$$\Pr(R \leq \mathbb{E}[R] + \lambda) \geq 1 - \frac{Var[R]}{Var[R] + \lambda^2}$$

If we let $C := 1 - \frac{Var[R]}{Var[R]+\lambda^2}$, then rearranging we get that $\lambda = \sqrt{\frac{Var[R]C}{1-C}}$. Thus for a variable $R$ that represents some sort of risk, and for some value of $0 < C < 1$, we can say

$$\Pr(R \leq P) \geq C$$

where $P := \mathbb{E}[R] + \sqrt{\frac{Var[R]C}{1-C}}$. In words, "there is at least $C$ chance that the risk is at most $P$." Alternatively, "we are at least $\frac{C}{100}\%$ confident that the risk is at most $P$."

## G APPENDIX G

To understand what exactly $\bar{R}_c^m(a)$ is an approximation of, consider instead calculating this risk using the true transition probabilities $t_a^{kj}$, We would get

$$r_k^0 := \mathbb{1}(q^k \text{ is observed and unsafe}) \tag{18}$$

$$r_k^{n+1}(a) := \begin{cases} 1 & \text{if } q^k \text{ is observed and unsafe} \\ \sum_{j=1}^N t_a^{kj} r_j^n & \text{otherwise} \end{cases} \tag{19}$$

$$r_k^{n+1} := \begin{cases} 1 & \text{if } q^k \text{ is observed and unsafe} \\ r_k^{n+1}\left(\arg\min_{a \in A} \bar{R}_k^{n+1}(a)\right) & \text{otherwise} \end{cases} \tag{20}$$

Note that we crucially still take the minimum risk action $a$ according to the agent's approximation $\bar{R}_k^{n+1}(a)$. In this case, the term $r_c^m(a)$ is the true probability of entering an unsafe state after selecting action $a$ in the agent's current state $q^c$ and thereafter selecting the actions that *the agent currently believes* will minimize the probability of entering an unsafe state over the horizon $m$. $\bar{R}_c^m(a)$ is the agent's approximation of $r_c^m(a)$.

We will later justify the use of $\bar{R}_c^m(a)$ as an approximation of $r_c^m(a)$, but for now let us consider why it makes sense to define $m$-step risk as $r_c^m(a)$. This because the action $a$ that minimizes believed risk is the action that the agent would choose if it was trying to behave as safely as possible, what I will call going into 'safety mode'. Consider the motivating example of a pilot learning to fly a remote control helicopter by incrementally expanding the set of actions they feels safe taking. They start by generating just enough lift to begin flying, then immediately land back down again. They repeat this process a few times until they feel that they have a good understanding of how the helicopter responds to this limited range of inputs. Then they take a risk (by either flying a bit higher, or attempting to move horizontally) and once again immediately land. As they repeat this process of taking small risks and landing to remain safe, they begin to expand their comfort zone. At some point after taking a risk, they will feel comfortable just coming back to a hovering position rather than landing, once they have become confident that they can hover safely. This suggests that a natural process for learning to operate in the face of risks is to repeatedly take small risks followed by going into safety mode until back in a confidently safe state. Thus, when calculating how risky an action is, it makes sense to consider the probability of entering an unsafe state given that after the action the agent will enter safety mode. $r_c^m(a)$ does exactly this.

As mentioned earlier, the other reason for defining the risk $r_c^m(a)$ in this way is that it makes it possible for the agent to attempt to calculate the risk without having to reason about the inter-dependency between the calculated risk and the agent's future actions. However, it does more than this. We will see in the next section that it in fact allows the agent to view $\bar{R}_c^m(a)$ as (an approximation of) the expected value of a random variable for the believed risk, where we can also approximate the *variance* of that random variable, allowing for deeper reasoning about action-selection for Safe RL.

