# OpenReview forum: "Risk-aware Bayesian RL for Cautious Exploration"
_ICLR.cc/2023/Conference — Submitted to ICLR 2023_

### Official Review · Reviewer_BCMN · 2022-10-21

**Confidence:** 4
**Correctness:** 2
**Technical Novelty And Significance:** 3
**Empirical Novelty And Significance:** 2
**Recommendation:** 3

**Clarity, Quality, Novelty And Reproducibility:**

**Clarity.** This paper is hard to follow due to inconsistency of mathematical notations. Also, related work section should be rewritten in an organized manner.

**Quality.** The quality of the theoretical analysis seems ok to me (though I am not fully confident), but that of experiments should be improved in terms of the selection of both benchmark problems and baselines.

**Novelty.** As far as I know, the proposed method is new. However, compared to recent work, I am not convinced whether the proposed method is better than existing ones. I don't think this paper has not supported the advantages of the proposed method neither empirically nor theoretically

**Reproducibility.** The source code is not attached and experimental conditions are not fully written in the paper (including appendix), I need to say that the reproducibility is low for now.

**Strength And Weaknesses:**

### Strength
**Problem Settings.** The problem settings are important and interesting.

**Theoretical Analysis** I think Theorem 3.1 is indeed a nice theorem though it is not clear to me how novel this theorem is.

### Weakness
**Paper writing.** First of all, the presentation of this paper should be improved in terms of notations. For example, $P$ or $Q$ are used in two meanings
+ P: Transition probability (in Section 2) and lowest risk level (in Section 3.3)
+ Q: finite set of states (in Section 2) and Q-function (in Section 3.4).

Such inconsistency makes this paper quite hard to follow. Also, the notations are far from standard as an RL paper, which gives the reader unnecessary and unessential burden. As an instance, I personally want the authors to avoid using $Q$ in other meanings than Q-function. State space is typically denoted as $S$ or $X$ in RL papers.

**Related work.** It would be better to write related work section in more organized manner. The current related work section is just a list of relevant papers, and it is hard to follow the story of the related work.

**Missing citations.** As a particularly relevant papers, I would like to list the following paper. I think it may be better to compare the authors' method with this existing work (the source code is released).
- As, Yarden, et al. "Constrained Policy Optimization via Bayesian World Models." International Conference on Learning Representations. 2021.

**Experiments.**  I don't think the experiment has been fully conducted. The benchmark task is very simple (i.e., grid world and PacMan), and the baseline methods are vanilla Q-learning. There are a lot of existing safe RL algorithms or benchmark tasks, so I think the authors should have compared with recent ones. For example, representatives of the benchmark tasks for safe RL is Safety-Gym or MuJoCo, and the notable safe RL baselines are PPO-Lagrangian (Here are just examples). At least, I do not think that the authors' claims are supported by the current experiments.

### Minor comments
- Since Moldovan and Abbeel (2012) is on safe exploration without ergodicity assumption, position of its citation seems weird to me. It would be more reasonable to cite it in the end of the next sentence.

**Summary Of The Paper:**

This paper studies safe reinforcement learning (RL) where safety violation must be bounded during training. They propose a new approach that balances the trade-off between efficient progress in exploration and safety guarantee. Specifically, the proposed approach updates Dirichlet-Categorical models of the state transition probabilities that describes the agent's behavior within the environment via Bayesian inference. They then approximate the agent's belief in terms of risks. They provide theoretical guarantees on the convergence on as well as empirically showing the performance of the proposed approach.

**Summary Of The Review:**

Though the problem setting and proposed method is interesting, I have several concerns as I listed in the Weakness. Hence, I recommend rejection for now.

---

> ### Author Response · Authors · 2022-11-19
> **Clarification on the questions**
>
> Thank you very much for the feedback. We comment on the weaknesses described in the review in the same order below:
> 1. Notations: we understand that the notation might not be conventional, and we’ll adapt the presentation of the article towards RL literature.
>
> 2. Missing relevant work: thank you for pointing out the related literature, we’ll certainly include it in our updated related work section.
>
> 3. Experiments: we acknowledge that the experiments are not indeed in continuous state or action, however, we would like to stress that the focus of this work is on introducing a new paradigm for safe exploration in RL and providing rigorous theoretical guarantees. Extending the presented ideas towards more scalable approaches and comparison to other baselines are indeed avenues of research we are currently pursuing.

---

> ### Comment · Area_Chair_tTnY · 2022-11-21
> **Any comments to the responses from authors?**
>
> Dear Reviewer BCMN,
>
> Thank you very much for your informative review.  The authors have provided responses to some of your concerns, but how did they change your evaluation?

---

### Official Review · Reviewer_Fqdj · 2022-10-26

**Confidence:** 5
**Correctness:** 2
**Technical Novelty And Significance:** 2
**Empirical Novelty And Significance:** 2
**Recommendation:** 3

**Clarity, Quality, Novelty And Reproducibility:**

The first point that should be addressed by the authors is the scalability of the proposed RCLR algorithm. Is it feasible the application of the RCLR algorithm to environments with large discrete state spaces or even continuous environments? Also, the complexity of the RCLR algorithm should be presented, and its connection to the hyper-parameter $m$ (back-propagation steps).

The main novelty of the proposed work should be also highlighted. Is the main contribution the keeping of a separate  Dirichlet-Categorical model for each environment state or the estimation of the risk associated with the agent’s behavior? In an abstract point, the way under which RCLR estimates the risk associated with the agent’s behavior can be seen as an MCTS.

As regards the clarity of the paper, some parts should be revised. Especially, the notation makes it hard for the reader to understand easily the idea behind the RCLR algorithm.

The empirical analysis is another limitation of this work. Specifically, it would be really interesting the examination of the RCLR algorithm on more challenging high-dimensional tasks. Also, comparisons should be conducted with more baselines apart from the vanilla Q-learning algorithm. The standard deviation should be also provided in Table 1. Finally, it is not clear the number of runs at PacMan (I guess that the presented results are coming by only one run).

**Strength And Weaknesses:**

**Strength**

- The problem of keeping agents safe during the training is of high interest and challenge. RCRL is able to estimate the risk associated with the agent’s behavior.
- A theoretical analysis is also provided.

**Weaknesses**
- The idea of keeping a separate  Dirichlet-Categorical model for each environment state is quite limited. The scalability of the RCLR algorithm is under question.
- Despite its merits the novelty of the RCLR algorithm is limited.
- Some parts of this work are hard to be followed by the reader, especially due to the notation.
- Empirical analysis is limited as experiments have been conducted in a small grid-world environment and on an over-simplified version of PacMan.

**Summary Of The Paper:**

This work introduces Risk-aware Cautious RL (RCRL) algorithm that allows agents to be trained and keep them safe. By keeping a Dirichlet-Categorical model for each environment state, PCRL approximates the risk associated with the agent’s behavior originating from local action selection. To evaluate the performance of the PCRL algorithm authors have conducted experiments at a small grid-world environment ($20 \times 20$) and a simplified version of PacMan (one ghost, 2 foods).


**Summary Of The Review:**

As aforementioned, the paper presented some interesting ideas but it has many weaknesses in its current version.

---

> ### Author Response · Authors · 2022-11-19
> **Clarification on the questions**
>
> Thank you very much for the feedback. We comment on the weaknesses described in the review in the same order below:
> 1. Novelty: to the best of our knowledge this is the first work that leverages the covariances of the risk terms in a Bayesian RL setting, which allows the agent to reason about the contribution of epistemic uncertainty to the risk level. As shown analytically and also empirically, this results in the agent making better informed decisions about how to stay safe during learning. Also, keeping a separate Dirichlet-Categorical model for each state-action pair allows the agent to estimate the risk at the highest granularity, and to leverage these models for multi-step lookaheads whenever needed.
>
> 2. Connection to MCTS: one of the advantages of RCRL, unlike MCTS-based approaches, is that it does not need to explore the sequence of outcomes until it reaches an unsafe state and only then backpropagate the risk. In fact, RCRL can be seen as a finite-horizon Bellman backup over the Dirichlet-Categorical models for risk estimation, which allows the agent to explore freely if an unsafe state is not in the horizon of the Bellman backup.
>
> 3. Scalability and high-dimensional experiments: we acknowledge that the experiments are not indeed in continuous state or action, however, we would like to stress that the focus of this work is on introducing a new paradigm for safe exploration in RL and providing rigorous theoretical guarantees. Extending the presented ideas towards more scalable approaches and complexity analysis are indeed avenues of research we are currently pursuing.
>
> 4. Table 1 and Pacman: many thanks for pointing out these issues, we’ll make sure to clearly state stds in table 1 and also the number of runs in Pacman.

---

> ### Comment · Area_Chair_tTnY · 2022-11-21
> **Any comments to the responses from authors?**
>
> Dear  Reviewer Fqdj,
>
> Thank you very much for your informative review.  The authors have provided responses to your concerns.  How did they change your evaluation, particularly on scalability, novelty, and empirical support?

---

### Official Review · Reviewer_HHjA · 2022-10-28

**Confidence:** 3
**Correctness:** 4
**Technical Novelty And Significance:** 4
**Empirical Novelty And Significance:** 4
**Recommendation:** 10

**Clarity, Quality, Novelty And Reproducibility:**

High clarity and quality. High novelty, as far as I can tell, but I might have missed some parts of the safe RL literature.
Most details is in the paper to reproduce, I think. But I strongly encourage the authors to release source code upon acceptance.

**Strength And Weaknesses:**

Strength:
* A proper probabilistic treatment on prior, safety and risk for RL given a Markov Decision Process.
* Tractable and useful approximations of expected risk (similar to traditional expected reward). Furthermore, a clear probabilistic safety requirement is used to bound the risk (using Cantelli Inequality), with a high interpretability.
* An excellent analysis and discussion on the results, and what safe RL entails.
* Good exposition of related works.
* The approach can be added on top of other RL methods (with discrete state action spaces?).

Weaknesses:
* No comparison to any other approach to safe RL. (However, given how formally well founded the presented approach is, this is more a valuable addition than a necessity.)
* Figure 2 (which I think is great and really important) can be improved:
  * I suggest that you draw red and yellow boundaries around where respective areas are in (b)-(d).
  * Make sure that all four figures have the same xtick and font size.
  * When referring to the figure, (y,x) is used which was confusing at first. E.g. the stand-still state in (b) is referred to as (13,1). I suppose that what is used is (row, column), but for figures/images (x,y) is standard.
  * For clarity, I suggest that you invert the yticks, without changing the figure, such that the starting location is (0,0). A $(-\hat{y},\hat{x})$ coordinate system for figures is only standard within UX/GUI/Graphics. It is better to use a $(\hat{x},\hat{y})$ coordinate system.
  * I understand that it does not fit, but I would have liked to have this figure in the main paper.
* The slippery PacMan experiment could be extended (different $P_\text{max}$, two ghosts, a little bit larger).

**Summary Of The Paper:**

The authors investigates a well founded probabilistic approach to safe RL, with primary contributions pertaining to safe training. Transition between states in the MDP is (during learning) modelled explicitly using a Dirichlet-Categorical model. Given a prior on transition probabilities (which can be fully informative) then the RL agent updates the belief of transition probabilities based on observed transition using Bayesian inference. The authors further derive suitable approximations of the risk associated with selecting action a in state q, based on the probability of transition and the cost of bad states. With the reasonable assumption that the agent can observe unsafe states m steps away (as transitions in the MDP), the approach is evaluated in slippery Gridworld and slippery PacMan showing impressive results. The paper ends with a well suited discussion on what the consequences for a learning agent when (proper) probabilistic safety guarantees are imposed, e.g. when the prior of the MDP transitions is weak.

**Summary Of The Review:**

The presented approach is (as far as I know) well suited in the literature, well founded, well presented, a highly suitable approach to safe RL (for training and otherwise) (in the applicable context of the approach) and sufficient empirical evaluation with interesting results which also motivate the method.

---

> ### Author Response · Authors · 2022-11-19
> **Clarification on the questions**
>
> Thank you very much for the feedback.
> We’ll certainly update Figure 2, and fix its xtick/ytick and coordinate reference issues. As for the Pacman experiment, we will be refreshing the final figures as well. Thank you also for the comment on the code base, at the moment we are polishing the code for the final release.

---

### Official Review · Reviewer_qaTz · 2022-11-03

**Confidence:** 3
**Correctness:** 3
**Technical Novelty And Significance:** 2
**Empirical Novelty And Significance:** Not applicable
**Recommendation:** 5

**Clarity, Quality, Novelty And Reproducibility:**

Clarity and Reproducibility: This paper is well-organized and easy to follow.

 Quality and Novelty: The novelty of the proposed method is unclear to me. Moreover, the experiments are insufficient.

**Strength And Weaknesses:**

Strengths:

1. The problem of safe exploration is important.

2. The proposed method is theoretically sound. The authors leverage a method from [1] to approximate the expected value and variance of the agent’s belief about the risk, and prove its convergence.

Weaknesses:

1. The novelty of the proposed method is unclear to me. Bayesian RL and risk-aware safe exploration have been widely studied in previous work. The authors may want to provide a detailed discussion on the novelty of the proposed method.

2. The experiments are insufficient. First, some important baselines are missing. The authors may want to compare to more baselines. Second, it would be more convincing if the authors could evaluate their proposed method on continuous control tasks, such as Mujoco [1].

[1] Todorov et al. "Mujoco: A physics engine for model-based control." international conference on intelligent robots and systems. IEEE, 2012.

**Summary Of The Paper:**

The authors propose a risk-aware bayesian reinforcement learning method to tackle the problem of safe exploration. Specifically, the authors assume that the agent maintains a Dirichlet-Categorical model of the MDP, and propose a method to derive an approximate bound on the confidence that the risk is below a certain level. Experiments in tabular environments demonstrate the effectiveness of the proposed method.

**Summary Of The Review:**

The proposed method is theoretically sound. However, the novelty of the proposed method is unclear to me, and the experiments are insufficient.

---

> ### Author Response · Authors · 2022-11-19
> **Clarification on the novelty and experiments**
>
> Thank you so much for the feedback. We comment on the weaknesses described in the review in the same order below:
>
> 1. Novelty: to the best of our knowledge this is the first work that leverages the covariances of the risk terms in a Bayesian RL setting, which allows the agent to reason about the contribution of epistemic uncertainty to the risk level. As shown analytically and also empirically, this results in the agent making better informed decisions about how to stay safe during learning. Also, the proposed method is versatile given that it can be added on to general RL training schemes, in order to maintain safety during learning.
>
> 2. Experiments: we acknowledge that the experiments are not indeed in continuous state or action, however, we would like to stress that the focus of this work is on introducing a new paradigm for safe exploration in RL and providing rigorous theoretical guarantees. Extending the presented ideas towards more scalable approaches and comparison to other baselines are indeed avenues of research we are currently pursuing.

---

> ### Comment · Area_Chair_tTnY · 2022-11-21
> **Any comments to the responses from authors**
>
> Dear Reviewer qaTz,
>
> Thank you very much for your informative review.  The authors have provided responses to your concerns.  How did they change your evaluation on the novelty and weaknesses in the experimental support?

---

### Official Review · Reviewer_G8rL · 2022-11-03

**Confidence:** 3
**Correctness:** 3
**Technical Novelty And Significance:** 2
**Empirical Novelty And Significance:** Not applicable
**Recommendation:** 3

**Clarity, Quality, Novelty And Reproducibility:**

**Clarity:**
The paper is well organized, but the presentation has minor details that could be improved, as discussed earlier in Strength And Weaknesses.

**Quality:**
The experimental evaluation is the bare minimum to support the main claims.

**Novelty:**
The main ideas of the paper have limited novelty.

**Reproducibility:**
The code is unavailable, which makes it difficult to reproduce the empirical results.

**Details Of Ethics Concerns:**

I do not find any ethical concerns.

**Strength And Weaknesses:**

**Strengths of paper:**
1. The problem of safety constraint violations in RL is important and has many real-world applications.

2. The authors empirically validated the performance of the proposed algorithm.

**Weakness of paper:**
1. The assumption of knowing which states are safe and which are unsafe (even locally) is very strong.

2. The motivational example is unsuitable for the considered problem setup in the paper as the state space can be very large or even continuous for the robotic problem.

3. The proposed approach will become computationally inefficient (or even infeasible) for a large number of states or actions (continuous state and action space).

4. It is unclear why authors do not use tighter concentration results for variance like Bernstein inequality.

5. No theoretical guarantee: No theoretical guarantee of how the proposed method will work compared to the optimal policy.


**Question and other comments.**

Please address the above weaknesses.

I have a few more questions/comments:
1. What is the initial belief used for transition probabilities?

2. Page 3, the line before Section 3: Change $0$ in $\alpha_a^{i0}$ to something else as it gives the impression that transition is happening from state $i$ to state $0$.

**Summary Of The Paper:**

This paper studies the safe reinforcement learning problem where safety constraint violations are bounded. Therefore, the agent's goal is to manage exploration and safety maintenance efficiently.

To achieve that, the authors propose a cautious RL scheme that uses Dirichlet-Categorical models of transition probabilities. They also empirically validate the different performance aspects of the proposed algorithm.

**Summary Of The Review:**

This paper significantly overlaps with my current work, and I am very knowledgeable about most of the topics covered by the paper.

---

> ### Author Response · Authors · 2022-11-19
> **Clarifications on the questions**
>
> Thank you very much for the feedback. We comment on the weaknesses described in the review in the same order below:
> 1. Assumption of knowing safe and unsafe states: Although this assumption only holds locally in this work, we can always add a probability distribution for safe/unsafe labels over the states, e.g. P_L: Q x {safe, unsafe} \rightarrow [0,1] such that for any state q \in Q we have \Simga_{label \in {safe,unsafe}} P_L (q, label) = 1. The risk calculation in a standard MDP can then be easily transferred to this label-probabilistic MDP setting by multiplying the original transition function P and P_L. We ensure to modify the paper accordingly.
>
> 2. Running example, and finite-state-action experiments: we acknowledge that the experiments are not indeed in continuous state or action, however, we would like to stress that the focus of this work is on introducing a new paradigm for safe exploration in RL and providing rigorous theoretical guarantees. Extending the presented ideas towards more scalable approaches is indeed an avenue of research we are currently pursuing.
>
> 3. Bernstein inequalities: In this work we used Cantelli’s inequality which is a special case of Chebyshev's inequality but it offers a sharper bound. Compared to other inequalities such as Bernstein’s inequality, Chebyshev's and Cantelli’s inequalities are more practical as they apply to normally distributed random variables. Bernstein’s inequality on the other hand, requires Bernoulli variables.
>
> 4. Theoretical guarantees: We would like to stress that the main objective of this work is to maintain safety for Cautious Exploration and we do have rigorous theoretical guarantees for ensuring this. The optimal policy in the sense of traditional RL, i.e. maximising the reward return, might not be safe in many scenarios as we described in the running example (and also Appendix C). We will make sure to emphasise this point further in the paper.
>
> Further points:
>
> 5. Initial belief: the details on priors can be found at the beginning of Section 4:
> Recall that a prior consists of a Dirichlet distribution p^i1_a , ..., p^iN-a \sim Dir(\alpha^i1_a,..., \alpha^iN_a ) for every state-action pair (q^i , a). We considered three priors:
>
> Prior 1 - completely uninformative: in this case we assigned a value of 1 to every \alpha. This yields a distribution that is uniform over its support.
>
> Prior 2 - weakly informative: we assigned a value of 12 to the α corresponding to moving in the correct direction, and a value of 1 to all other \alpha’s. This gives a distribution in between Prior 1 and Prior 3 in both degree of bias and concentration.
>
> Prior 3 - highly informative: we assigned a value of 96 to the α corresponding to moving in the correct direction, and a value of 1 to all other \alpha’s. This gives a distribution that is highly concentrated, and for which the mean values of the transition probability random variables are the true transition probabilities of the MDP, and hence unbiased.
>
> 6. Line before Section 3: thanks for the comment, we’ll ensure to change it accordingly.

---

> > ### Comment · Reviewer_G8rL · 2022-12-07
> > **Thank you for response**
> >
> > Hi,
> >
> > Thank you for your response! The assumption of knowing safe and unsafe states even locally is enough to discard the actions which otherwise lead to unsafe states. Further, we can use the Bernstein inequality for any IID sequence of bounded random variables, not only Bernoulli random variables (check Audibert et al., 'Exploration–exploitation tradeoff using variance estimates in multi-armed bandits' for reference). After reading other reviews and your responses, I will keep my recommendation unchanged.

---

> ### Comment · Area_Chair_tTnY · 2022-11-21
> **Any comments to the responses from authors?**
>
> Dear Reviewer G8rL,
>
> Thank you very much for your informative review.  The authors have responded to your concerns.  How did they change your evaluation on each of the weaknesses that you pointed out?

---

### Decision · Program_Chairs · 2023-01-20

**Decision:**

Reject

**Justification For Why Not Higher Score:**

The proposed method has significant limitation in its applicability, and it is unclear how it can be applied beyond toy tasks in tabular environments.

**Justification For Why Not Lower Score:**

N/A

**Metareview: Summary, Strengths And Weaknesses:**

This paper proposes a safe reinforcement learning (RL) method, where the agent maintains Dirichlet-Categorical models of state transitions, and the proposed method of approximating the confidence that the risk is below a certain level is used to control exploration.  The safe RL is certainly important, and the paper proposes a theoretically well grounded method for this important subject, which is the main strength of the paper.

The major weakness pointed out by reviewers is its scalability.  While the effectiveness of the proposed method is empirically demonstrated in tabular environments, it is not at all clear how the proposed method can be applied with functional approximations, which are needed for most practical tasks.